# Reversible HuR-microRNA binding controls extracellular export of miR-122 and augments stress response

Kamalika Mukherjee[1], Bartika Ghoshal[1], Souvik Ghosh[1,†], Yogaditya Chakrabarty[1], Shivaprasad Shwetha[2], Saumitra Das[2] & Suvendra N Bhattacharyya[1,*]

## Abstract

microRNAs (miRNAs), the tiny but stable regulatory RNAs in metazoan cells, can undergo selective turnover in presence of specific internal and external cues to control cellular response against the changing environment. We have observed reduction in cellular miR-122 content, due to their accelerated extracellular export in human hepatic cells starved for small metabolites including amino acids. In this context, a new role of human ELAV protein HuR has been identified. HuR, a negative regulator of miRNA function, accelerates extracellular vesicle (EV)-mediated export of miRNAs in human cells. In stressed cells, HuR replaces miRNPs from target messages and is both necessary and sufficient for the extracellular export of corresponding miRNAs. HuR could reversibly bind miRNAs to replace them from Ago2 and subsequently itself gets freed from bound miRNAs upon ubiquitination. The ubiquitinated form of HuR is predominantly associated with multivesicular bodies (MVB) where HuR-unbound miRNAs also reside. These MVB-associated pool of miRNAs get exported out via EVs thereby delimiting cellular miR-122 level during starvation. Therefore, by modulating extracellular export of miR-122, HuR could control stress response in starved human hepatic cells.

Keywords Ago; extracellular vesicles; HuR; miRNA export; miRNAs; ubiquitination

Subject Categories Membrane & Intracellular Transport; Post-translational Modifications, Proteolysis & Proteomics; RNA Biology

## Introduction

miRNAs, the 22-nucleotide-long noncoding RNAs, form miRNP complexes with Argonaute (AGO) proteins and regulate majority of genes by imperfect base pairing to the 3′UTR of target messages [1].

The majority of biochemical pathways in humans are miRNA controlled, and human diseases including several forms of cancer are associated with abnormal expression of miRNAs [2–4].

miRNPs repress target genes either by inhibiting the translation or by affecting the stability of the target mRNAs [5,6]. Tissue-specific expression and activities of hundreds of miRNAs are controlled at different levels to fine tune the gene expression in metazoan animals [7]. Expressions and activities of miRNAs are also sensitive to cellular context [8–10]. Despite being an important regulator of gene expression in eukaryotes, factors controlling miRNA stability in mammalian cells have remained largely unidentified.

Examples of both positive and negative feedback mechanisms to regulate miRNA level and activity mostly remain limited to specific miRNAs and are largely operative at miRNA synthesis and maturation steps [11]. miRNAs are not only contained within cells but can also be found in various body fluids including peripheral blood plasma, saliva, serum, and milk [12]. These extracellular miRNAs are protected from degradation either by packaging in microvesicles (MVs) such as apoptotic bodies, shedding vesicles, and exosomes or even as part of free Ago proteins–miRNA complexes [13–15]. Some miRNA species were also found in purified fractions of high-density lipoprotein (HDL) from human plasma [16,17]. Reports indicate that these secreted extracellular miRNAs, especially those within the membrane bound microvesicles, are internalized by other cells and alter gene expression and mediate functional effects in the recipient cells [18–24]. Exosome-mediated delivery can also compensate the loss of miRNAs in neighboring cells [25]. Recent report suggests stability of miRNAs may also be controlled by extracellular export where impaired exosomal export in growth-retarded mammalian cells increases cellular miRNA level [10].

miRNAs get regulated by stress but themselves are also the key controllers of stress response [26]. In recent times, several reports have been published on the connection of miRNAs with stress in animal cells [27,28]. Human ELAV protein HuR was identified earlier as a derepressor of miRNA function that inhibits the action of miRNA on *cis*-bound target mRNAs primarily by uncoupling the miRNPs from the target messages and mobilizing them from P-bodies in amino acid-starved human hepatoma cells [29]. HuR is

---

1  RNA Biology Research Laboratory, Molecular Genetics Division, CSIR-Indian Institute of Chemical Biology, Kolkata, India
2  Department of Microbiology and Cell Biology, Indian Institute of Science, Bangalore, India
   *Corresponding author. Tel: +91 33 24995783; Fax:+91 33 24735197; E-mail: sb@csiriicb.in
   †Present address: Biozentrum, University of Basel, Basel, Switzerland

a stabilizer of AU-rich element (ARE) containing mRNAs [30], but it is unknown how HuR, by preventing miRNA–target mRNA interaction, may affect miRNA stability in animal cells.

HuR-regulated export of miR-122 from the human hepatic cells, starved for different metabolites including amino acids, is the prime focus of this article. We have identified an accelerated extracellular export of miR-122 that reduces its cellular level in Starved human hepatic cell Huh7. HuR protein acts as a regulator of the export process of miRNAs. HuR replaces miRNPs from co-targeted messages and itself binds with miRNAs to ensure miRNA unloading of Ago2 protein. Similarly, ubiquitination of HuR causes unbinding and export of HuR-bound miRNAs. Impairment of stress-induced export of miR-122 due to HuR depletion was found to be associated with poor stress response and autophagy in Starved hepatic cells. Therefore, by regulating extracellular export of miRNAs, metabolic status of the cells can be changed.

## Results

### Extracellular export of miR-122 in Starved human hepatoma cells

Amino acid starvation-induced derepression of miR-122 target genes has been reported previously in human hepatic cells. In Starved cells, RNA-binding protein HuR, by binding with the AU-rich elements on the 3′UTR of its targets, replaces the *cis*-bound miRNPs and ensures a reversal of miRNA-mediated repression of specific sets of genes [29]. To understand the fate of replaced miRNPs, we measured the miR-122 content in Fed and Starved cells and noticed a reduction in miR-122 content upon starvation along with an enhancement in expression of miR-122 target gene CAT-1 (Fig 1A and B). We quantified other miRNAs in Starved cells and observed downregulation was specific for miR-122 as cellular levels of other miRNAs did not show significant downregulation upon starvation (Fig 1C). Extracellular vesicle (EV)-mediated export is reported to be a key mechanism for lowering cellular miRNA content in human cells [10]. Huh7 cells release EVs (Fig EV1) [25] and measurement of miRNA content of EVs revealed an increase in miR-122 level in EVs derived from Starved Huh7 cells than Fed cell control. The levels of miR-24 measured in EVs from Fed and Starved cells were comparable (Fig 1C). The increase in mature miR-122 content in EVs upon starvation was not reflected in the changes of pre-miR-122 levels (Fig 1C). Starvation-induced changes in mature miR-122 content both in the EV-associated and in cellular miRNA pools were not accompanied by an alteration in the length of miR-122 in Starved cells or released EVs (Fig 1A and B). Starvation did not show any significant effect on the levels of EVs released as marker proteins CD63 or HRS showed similar levels in EVs derived from Fed and

---

**Figure 1.   Starvation induces extracellular export of miR-122 in mammalian hepatic cells.**

A, B   Levels of miR-122 in Huh7 cells and released EVs either untreated (Fed) or subjected to starvation for metabolites including amino acids for 16 h (Starved). miR-122 signals were detected by Northern blotting and position of the $^{32}$P-labeled oligos that served as size markers is shown in the M lane (A). U6 snRNA was used as loading control. Cellular CAT-1 levels were measured by qRT–PCR using GAPDH mRNA as control (B) (mean ± s.e.m., n = 3). A scheme of experiment is shown in the top panel in (B).

C, D   Cellular and extracellular levels of different miRNAs and EV-associated proteins in Fed and Starved Huh7 cells. Relative changes in cellular and extracellular levels (EV-associated) of miR-122 and miR-24 in Fed or 8 h Starved Huh7 cells were quantified by qRT–PCR and plotted (mean ± s.e.m., n = 3). Levels of individual miRNAs in Fed condition were taken as unit. Fold change (with SD) in the cellular level of five different miRNAs and pre-miR-122 measured is shown in the bottom panel (C). Cellular miRNA levels were normalized against U6 snRNA. Western blot analysis of different exosomal proteins and Ago2 in EVs isolated under Fed or Starved conditions. CD63 is an exosomal marker protein and levels of HuR in EVs were quantified against respective CD63 levels. Mean of three independent measurements is plotted (middle panel in D). Calnexin and GAPDH are not detected in EVs but visible in the cellular extract. EV-associated HA-HuR levels for Fed and Starved Huh7 cells expressing HA-HuR were also detected in the bottom panel in (D).

E   Effect of GW4869 treatment on cellular miRNA content in Fed and Starved cells. Levels of miRNAs were measured by real-time quantification and normalized against U6 snRNA. Mean data are from three independent experiments. DMSO treatment was used as control for GW4869-treated cells (mean ± s.e.m., n = 3).

F   CAT-1 and aldolase mRNA expression in DMSO- or GW4869-treated Starved Huh7 cells. qRT–PCR technique was adopted for quantification using GAPDH mRNA values for normalization (mean ± s.e.m., n = 3).

G   Effect of siSMPD2 treatment on miR-122 content of EVs from Fed and Starved Huh7 cells. EVs from Fed and Starved Huh7 cells, depleted for SMPD2, were isolated, and miR-122 content was measured by qRT–PCR and normalized against protein content of EVs. Percent change in EV-associated miR-122 levels upon starvation both for control and for siSMPD2-treated cells is shown above the respective bars (mean ± s.e.m., n = 4).

H   Effect of siSMPD2 treatment on CAT-1 mRNA content (left panel) and miR-122 levels (right panel) in Huh7 cells. RNAs from Fed and Starved Huh7 cells, depleted for SMPD2, were isolated and miR-122 and CAT-1 mRNA contents were measured by qRT–PCR and normalized against U6 snRNA. GAPDH mRNA was used for normalization of CAT-1 mRNA levels (mean ± s.e.m., n = 3). miR-122 levels for siCon-treated cells (Fed and Starved) were considered as units.

I, J   A schematic representation of the experiment to separate extracellular vesicles (EVs) on OptiPrep™ density gradient (left panel). Densities of fractions 1–10 are plotted and a best-fit curve is drawn (I; right upper panel). CD63 levels in individual fractions of Fed and Starved cells were detected by Western blots to confirm the presence of exosomes of Fed and Starved cells (I; right lower panel). Mean $C_t$ values of miR-122 in RNAs isolated from exosome-enriched (8–9) vs non-exosomal (1–3) fractions were analyzed and plotted (J) (mean ± s.e.m., n = 3).

K   Effect of starvation on apoptotic cell numbers of Huh7 cells. Western blot detection of apoptosis marker cytochrome c in cellular and in EV-associated fractions of Fed and Starved Huh7 cells. β-Actin and CD63 were used as loading controls for cellular and EV-associated fractions, respectively (left panel). TUNEL-positive cells in Fed and Starved Huh7 cells. TUNEL assays of Fed and Starved Huh7 cells were performed, and TUNEL-positive cells were quantified (mean ± s.e.m., n = 60). Fed, but DNase-treated, cells were used as positive control (+ve control) (right panel).

L   Effect of thapsigargin (TG) treatment of Huh7 cells on cellular and EV-associated miR-122 levels. A schematic representation of the experiment (upper panel). miR-122 levels were measured by qRT–PCR in total cellular RNA and in EVs released by the TG treated cells. Values were normalized either against U6 snRNA or protein content of EVs (mean ± s.e.m., n = 3) (lower panels) for cellular and EV-associated RNA, respectively.

M   Effect of TG on cellular level of eIF2-α and its phosphorylated form, Ago2 and CD63 in EVs. β-Actin was used as loading control for cellular samples.

Data information: For statistical analysis, all experiments were done minimum three times and P-values were calculated. ns: non-significant, *P < 0.05, **P < 0.01, ***P < 0.0001. P-values were determined by paired t-test. Positions of size markers in protein gels used for respective Western blot analysis are shown against each panel. For estimations of relative level of either miRNAs or mRNAs, $C_t$ values only within the range of 20–32 were considered for analysis.

Source data are available online for this figure.

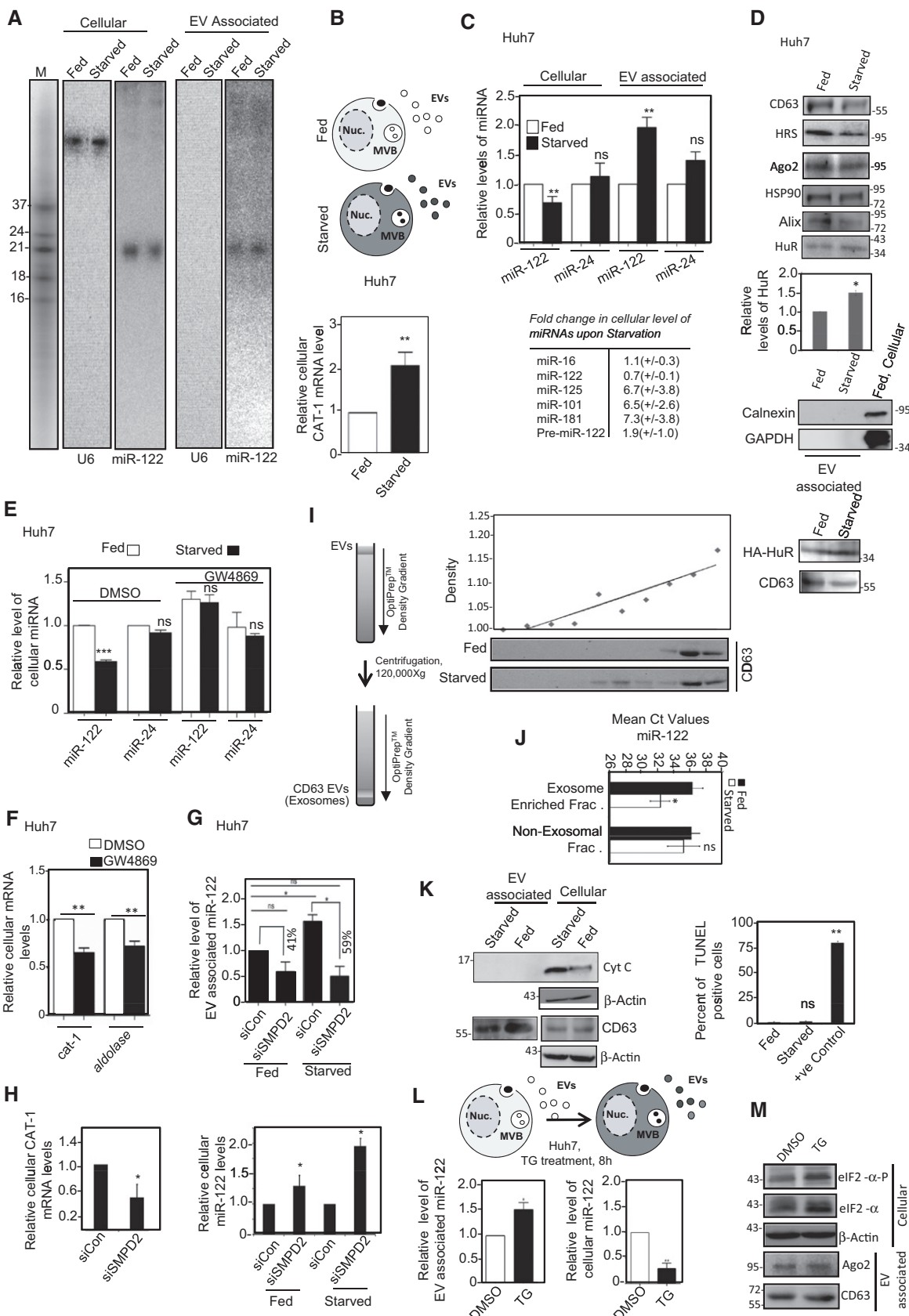

Figure 1.

Starved cells. The level of Alix drops in EVs released by Starved cells, whereas levels of HSP90 did not change as depicted in Western blots done with EVs derived from Fed or Starved cell (Fig 1D). Exosomes constitute the major fraction of EVs and GW4869 specifically blocks exosomal miRNA export in mammalian cells by inhibiting neutral sphingomyelinase II (SMPD2) [21]. GW4869 could prevent starvation-induced decrease in miR-122 in Huh7 cells and lower starvation-related increase in miR-122 target CAT-1 and aldolase mRNA in Starved cells (Fig 1E and F). RNAi-mediated depletion of SMPD2, the target of GW4869, also lowered miR-122 content of EVs derived from Fed and Starved Huh7 cells, and prevented starvation-induced enhancement in miR-122 export (Fig 1G). This was accompanied by increase in cellular miR-122 and lowering of target *Cat-1* in siSMPD2-treated cells (Fig 1H). Confirming the exosomal nature of the EVs associated with increased miR-122 released by Starved Huh7 cells, only the exosomal CD63-positive fractions of EVs, resolved on an OptiPrep™ density gradient, showed enhancement in its miR-122 content upon starvation (Fig 1I and J). The contribution of apoptotic bodies that may constitute a large pool of extracellular RNA in any given cell culture medium was also tested. Presence of apoptotic bodies should be indicated by presence of cytochrome c, but no cytochrome c was detected in EVs released either by Fed or by Starved Huh7 cells. There was also no major change in the apoptotic cell numbers during starvation as evident from TUNEL assays (Figs 1K and EV1F). Enhanced EV-mediated export of miRNA not only happens under metabolic stress but was also detected in cells treated with thapsigargin (TG), an inducer of ER stress. TG-induced stress lowered cellular miR-122 level with an increase in EV-associated miR-122 levels (Fig 1L). It happens without a change in the content of Ago2 or CD63 proteins in EVs (Fig 1M). Therefore, in nutrient Starved or in TG-treated Huh7 cells, enhanced extracellular export of miR-122 has been documented and released miR-122 in culture supernatant was found to be associated with CD63-positive EVs.

## miRNA binding of HuR get enhanced in Starved hepatic cells and HuR is necessary for miR-122 export and stress response

HuR is essential for starvation-induced relief of miRNA-mediated repression of target messages in Huh7 cells. Under starvation, HuR relocalizes to the cytoplasm from the nucleus and its binding with the target messages prevents binding of miRNPs to co-targeted messages [29]. Interestingly, we observed a relative increase in the content of HuR in EVs upon starvation (Fig 1D). HuR binding to miRNAs has been reported earlier where selective binding of HuR for specific miRNAs has been documented [31]. We immunoprecipitated HuR both from Fed and Starved Huh7 cells and documented an enhanced binding of HuR with miR-122 in Starved cells. Consistent with previously published data, we also found enhanced HuR binding of miR-21 in Starved cells (Fig 2A) [32]. Increase in the levels of both HuR and miR-122 in the EVs derived from the culture supernatant of Starved cells suggests a possible role of HuR in EV-mediated export of miRNAs. HuR, as a reliever of miRNA function on ARE-containing mRNAs, mobilizes miRNA-repressed mRNAs out of P-bodies to induce protein synthesis in Starved cells [29]. Does HuR have a role in controlling EV-mediated export of miRNAs and hence miRNA activity in Huh7 cells? Depletion of HuR by RNAi in Huh7 cells resulted reduction in the EV content of miR-122 which was consistent with an increase in cellular miR-122 pool and decrease in levels of miR-122 targets in the HuR-depleted cells. Two other miRNAs tested also showed similar trend, that is, a reduction in EV-associated miRNA levels upon HuR depletion (Fig 2B and C).

Does impaired miR-122 export affect stress response in Starved cells? To test the role of HuR-promoted export of miRNA in controlling stress response in Huh7 cells, we measured the effect of depletion of HuR on miR-122 level in Starved cells. As expected, we observed substantial increase in miR-122 content in HuR-depleted Huh7 cells. This was consistent with a decrease in EV-associated miR-122 level (Fig 2D). With RNAi-mediated reduction in cellular

**Figure 2.  Human ELAV protein HuR binds miRNA in Starved cells and is necessary for extracellular export of miRNA.**

A    Relative amount of miRNAs bound to HuR that are immunoprecipitated either from Fed or from Starved Huh7 cells. Immunoprecipitated materials were separated into two equal halves to do Western blot analysis of HuR, and relative quantification of associated miR-122 and miR-21 was done by quantitative RT–PCR (mean ± s.e.m., n = 3). GFP used as negative control to confer the specificity of the anti-HuR 3A2 antibody used in immunoprecipitating HuR. The miR-122 content in the GFP antibody-immunoprecipitated materials was too low to get detected reliably using the qRT–PCR.

B, C  Effect of HuR depletion on cellular and EV content of miRNAs in Huh7 cells. Schemes of HuR depletion experiments in Huh7 cells (B). Relative levels of miRNAs were measured in EVs released from siCon- or siHuR-treated cells by qRT–PCR. Cellular miRNA contents were normalized against U6 snRNA. Levels of each miRNA in control siRNA (siCon)-treated set were taken as unit. In siHuR-treated cells, depletion of HuR was monitored by Western blotting (mean ± s.e.m., n = 4) (B). CAT-1 and aldolase mRNA levels of siHuR- and siCon-treated Fed Huh7 cells were quantified by qRT–PCR and normalized against GAPDH mRNA (C).

D    Effect of HuR knockdown on cellular level of different miRNAs in Starved Huh7 cells. Relative levels were estimated by real-time qPCR, normalized with respect to U6 snRNA, and plotted in the left panel. Levels of miR-122 in EVs of Starved siHuR- and siCon-treated Huh7 cells were also measured by RT–PCR and plotted in the right panel (mean ± s.e.m., n = 3).

E    Levels of eIF-2α and its phosphorylated form, estimated by Western blot using specific antibodies, in Starved siCon- or siHuR-treated Huh7 cell extracts. Western blot data of HuR confirm reduction of this protein upon siRNA treatment. CAT-1 and aldolase mRNA levels were estimated by real-time quantification and normalized against GAPDH level. Values of siCon-treated samples were taken as unit (mean ± s.e.m., n = 3). In the right panel, the relative quantity of phosphorylated eIF-2α measured by densitometry was plotted. Levels of p38 and phospho-p38 levels in siCon- and siHuR-treated Starved Huh7 cells were detected by Western blot.

F    Western blot analysis to detect the level of phosphorylated eIF-2α and eIF-4E-BP1 in lysates of Starved and HuR-depleted Huh7 cells either transfected with anti-let-7a or anti-miR-122 oligos. miR-122 inactivation was monitored by measuring CAT-1 mRNA level by qRT–PCR using GAPDH mRNA as control (mean ± s.e.m., n = 3). * denotes the phosphorylated eIF-4E-BP1. β-Actin was used as loading control. Relative levels of phosphorylated eIF-2α were measured by densitometric estimation of three independent measurements.

Data information: ns: non-significant, *P < 0.05, **P < 0.01. P-values were determined by paired t-test. Positions of size markers in protein gels used for respective Western blot analysis are shown against each panel. HC: heavy chain and LC: light chain of IgG used for immunoprecipitation reaction. For estimations of relative level of either miRNAs or mRNAs, $C_t$ values only within the range of 20–32 were considered for analysis.

HuR level, we did not find an effect on cellular miR-21 level although miR-16 levels showed an increase. Reduction in EV-associated miR-122 was accompanied by reduced expression of miR-122 target genes and reduced eIF2-α phosphorylation, a hallmark of stress response under amino acid starvation [33] that was also accompanied by a reduction in phospho-p38 level (Fig 2E). Does this impaired stress response happen specifically because of elevated miR-122 level in HuR-depleted Huh7 cells? To find it out,

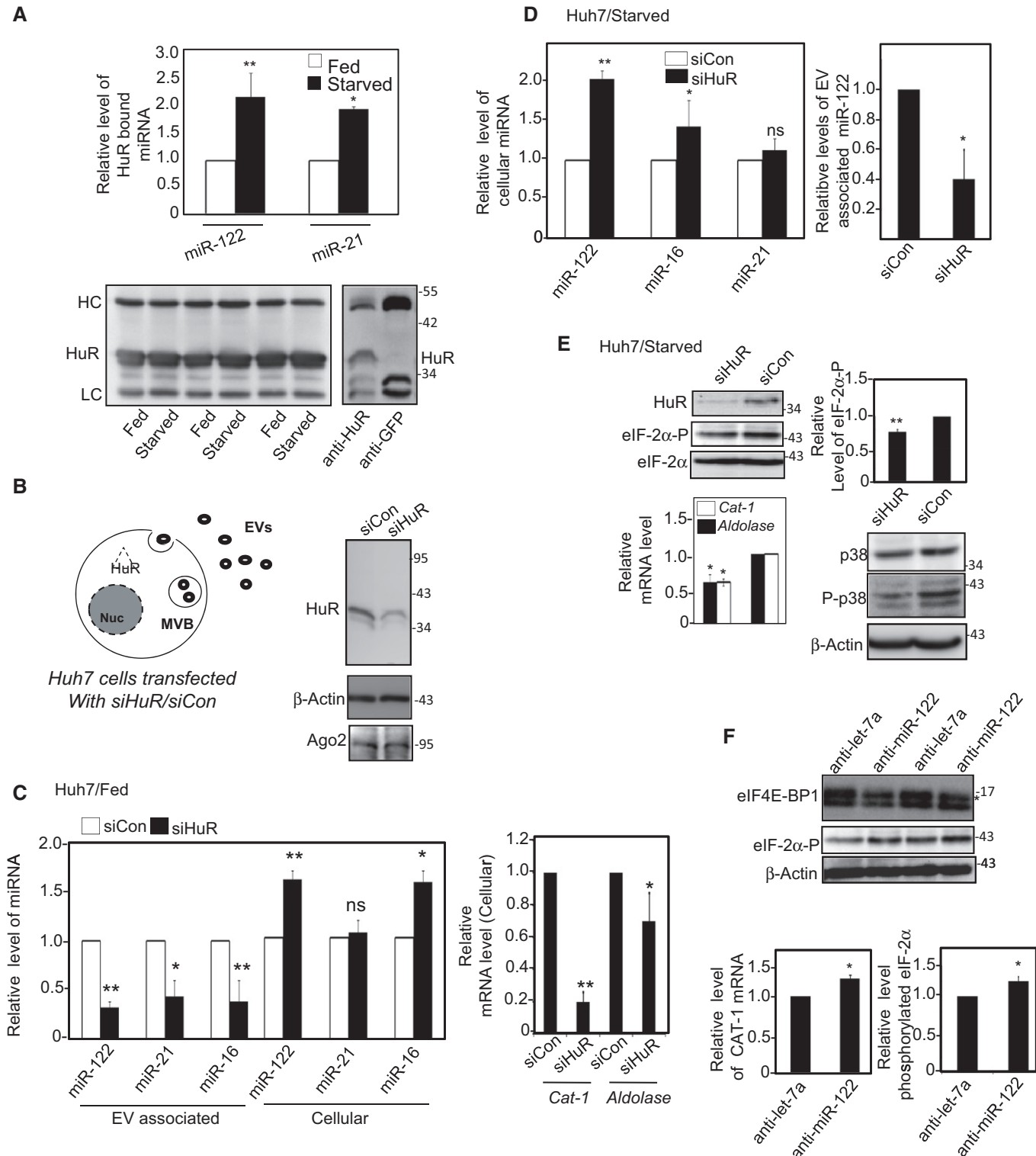

**Figure 2.**

we introduced anti-miR-122 oligos in Huh7 cells. Introduction of anti-miR-122 oligos inactivates the excess miR-122 accumulated due to impaired HuR-mediated export of miR-122 and restored the phosphorylation of eIF2-α. We also detected a loss in phosphorylation of eIF-4EBP1 in anti-miR-122-treated cells compared to control anti-let-7a-treated cells. Loss of phosphorylation of eIF4E-BP1 and increased eIF-2α phosphorylation suggests slowdown of protein translation process, as a part of restored stress response. Inactivation of accumulated miR-122 in HuR-depleted Starved Huh7 cells by anti-miR-122 oligos was responsible for the restoration of stress response. It was accompanied by an increase in CAT-1 mRNA level, confirming reduced miR-122 activity in anti-miR-122-treated cells, compared to the control anti-let-7a-treated cells (Fig 2F, lower panel). These results support the idea that the effect of HuR depletion on stress response of Huh7 cells is due to enhanced cellular miR-122 level and miR-122 activity reduction is required for an augmented stress response in Starved Huh7 cells. Starvation-related stress is usually coupled with increased autophagy [34]. Interestingly, such increase in autophagy was not detected in Starved Huh7 cells (Fig EV2A). Depletion of HuR reduces expression of LC3BII and p62 levels in Starved cells and was consistent with reduced phospho-p38 (Fig 2E) and increased phospho-mTOR levels in HuR-depleted cells (Fig EV2B). To test the contribution of increased miR-122 for reduced expression of autophagy markers in siHuR-treated Huh7 cells, we inactivated miR-122 and detected reduced phospho-mTOR and increased p62 expression in Starved Huh7 cells treated with anti-miR-122 oligos compared to anti-let-7a oligo-treated control cells (Fig EV2C). Therefore, in Starved Huh7 cells, miR-122, possibly by activating mTOR and inhibiting eIF2-alpha phosphorylation via p38 pathway activation, restricts both stress response and autophagy. HuR antagonizes this inhibition of autophagy by miR-122 possibly by exporting it out via EVs to augment autophagy and stress response in Huh7 cells Starved for amino acids and metabolites (Fig EV2D).

## HuR-driven export of miR-122 from mammalian liver cells

Excess HuR, known to stabilize and promote translation of mRNAs with ARE element in their UTRs [35], also enhances miRNA content of EVs and lowers cellular levels of corresponding miRNAs in Huh7 cells (Fig 3A–C). Although there was no increase in the marker protein CD63 or Alix levels in EVs isolated from HA-HuR-expressing cells compared to control group of cells, there was an increase in the content of HuR in the EVs isolated from HA-HuR-expressing cells. Importantly, HA-HuR expression specifically alters miR-122 level in EVs without affecting the miR-122 content of non-EV fractions (Fig 3E). Similarly affinity-purified CD63-positive EVs from HA-HuR-expressing cells also showed enhanced miR-122 content compared to EVs from pCIneo-transfected control cells (Fig 3D and F). HuR-mediated export of miRNAs was blocked by NSMase inhibitor GW4869 that further suggests that the enhanced extracellular export of miRNA occurs via exosomal/EV pathway (Fig 3G).

Specific association of enhanced extracellular miRNAs with EVs from the HA-HuR-expressing cells was further confirmed when CD63-positive EVs (exosomes) from Huh7 cells expressing HA-HuR were found to get specifically enriched for exported miR-122 (Fig 3H). HA-HuR expression did not induce any increase in apoptotic cell numbers or release of cytochrome c-positive apoptotic bodies that could potentially contribute to this increase in extracellular miR-122 content of HA-HuR-expressing Huh7 cells (Fig 3I). We documented a decrease in exosomal and cellular CD63 content upon HA-HuR expression, but the reason of this drop was not apparent.

---

**Figure 3.  HuR is sufficient for EV-mediated export of miRNAs.**

A   Northern blot of miR-122 in Huh7 cells expressing the control and HA-HuR expression plasmids. The U6 snRNA was used for loading control. Position of the [32]P-labeled oligos, blotted to the membrane used for Northern blotting, served as size markers and are shown in the M lane.

B   Schematic representation of experiments done in Huh7 cells expressing HA-HuR (bottom). For control, pCIneo-transfected Huh7 cells were used. Relative expression of HuR in transfected cells was analyzed by Western blot (top). β-Actin blot was used as loading control.

C   Cellular and EV-associated miRNA levels were measured in Huh7 cells either expressing the pCIneo or HA-HuR-expressing vectors by qRT–PCR (mean ± s.e.m., *n* = 5). Cellular miRNA levels were normalized against U6 snRNA.

D   Experimental scheme for EV isolation either by affinity purification technique using CD63-specific biotinylated beads or by ultracentrifugation from Huh7 cell-conditioned medium (Huh7CM).

E   Effect of HA-HuR expression on miRNA levels of EV and non-EV fractions of the culture supernatant collected from control and HA-HuR-encoding plasmid-transfected Huh7 cells. For isolation of non-EV fraction, supernatant obtained after removal of EVs by ultracentrifugation was used for RNA preparation (left panel) (mean ± s.e.m., *n* = 3). Effect of HuR expression on CD63, Alix, and HuR levels in EVs. Extracts of EVs isolated from control or HA-HuR-expressing Huh7 cells were Western blotted for respective proteins (right panel).

F   Relative miR-122 levels in CD63-positive affinity-purified EVs isolated from HA-HuR- and pCIneo-transfected Huh7 cells were quantified by qRT–PCR (mean ± s.e.m., *n* = 3). The levels of miR-122 were normalized against CD63 content of EVs and normalized vales with SD are shown below the bars.

G   Effect of GW4869 on EV-associated content of three different miRNAs in HA-HuR-expressing Huh7 cells quantified by qRT–PCR (mean ± s.e.m., *n* = 5). EVs from DMSO-treated cells were used as control.

H   A schematic representation of the experiment done to characterize EVs isolated from Huh7 cells expressing HA-HuR or pCIneo, respectively, by running OptiPrep™ density gradient (left panel) and Western blotting for CD63 (right upper panel). Relative miR-122 levels in fractions 1–2 and 8–9 were compared by qRT–PCR (right lower panel).

I   Levels of apoptotic cells in HA-HuR-expressing Huh7 cells. Western blot detection of apoptosis marker cytochrome c was done (top) for EV-associated and cellular fractions of HA-HuR- and pCIneo (control)-expressing Huh7 cells, respectively. Western blot for HA-HuR was done to analyze the expression of HA-HuR in EVs isolated from HA-HuR- and pCIneo-expressing Huh7 cells. CD63 and β-actin were used as loading controls for EV-associated and cellular fractions, respectively (upper panels). TUNEL assay of HA-HuR- and pCIneo-expressing and non-transfected Huh7 cells was performed and representative pictures of HA-HuR- and pCIneo-expressing cells along with picture of non-transfected, but DNase-treated (positive control) cells were compared (lower panels). Scale bar, 50 μm.

Data information: ns: non-significant, *P < 0.05, **P < 0.01, ***P < 0.0001. *P*-values were determined by paired *t*-test. Positions of size markers in protein gels used for respective Western blot analysis are shown against each panel. For estimations of relative level of either miRNAs or mRNAs, $C_t$ values only within the range of 20–32 were considered for analysis.

Interestingly, the HuR-driven export of miRNA happens irrespective of cell types. Depletion of HuR curtailed the content of let-7a in the EVs released by siHuR-treated human breast cancer cell

MDA-MB-231 with concomitant increase in cellular let-7a level (Fig EV3A). Importantly, miRNA also increased in cells treated with GW4869, the blocker of exosomal export of miRNAs [10]. GW4869

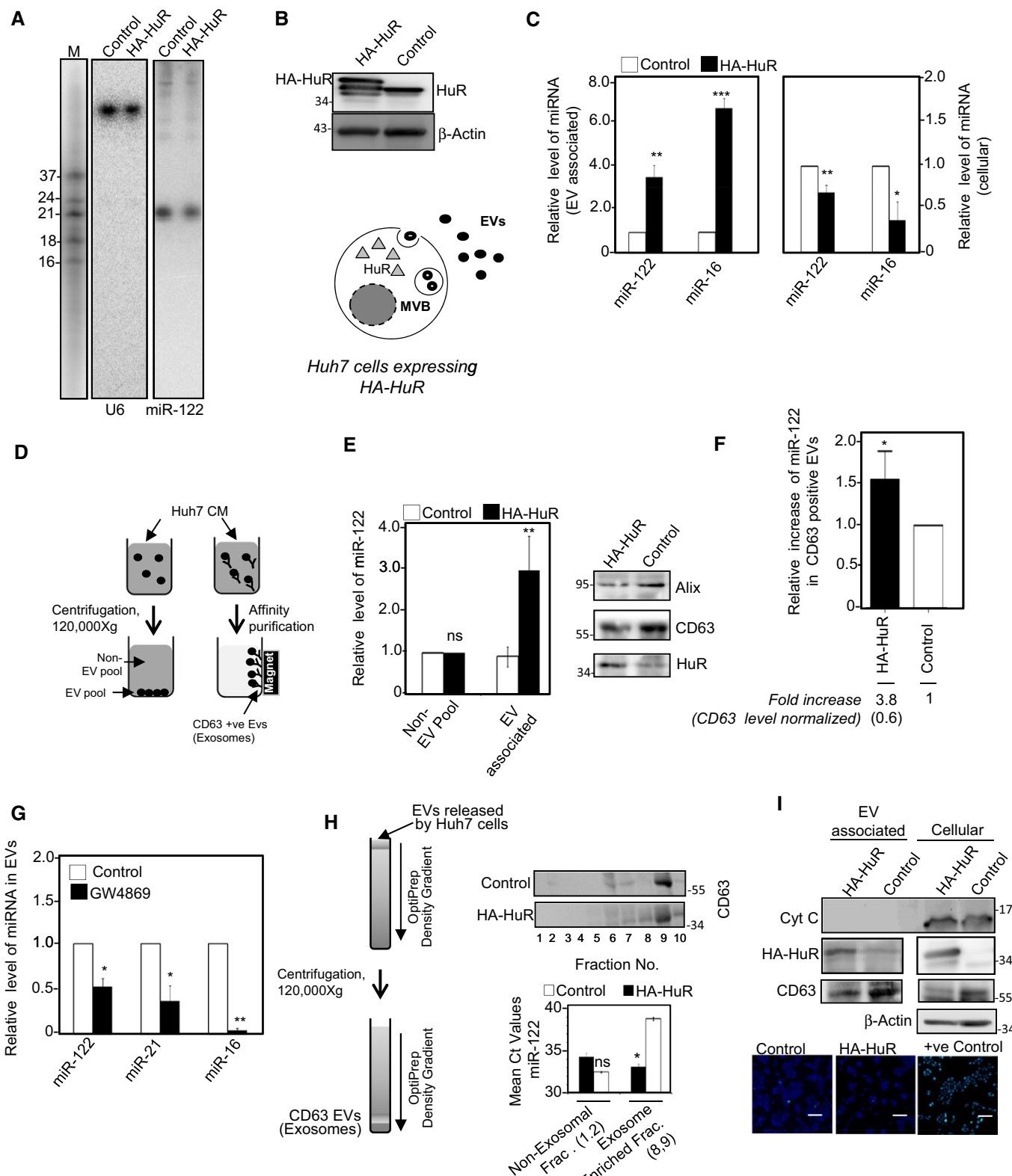

**Figure 3.**

treatment increased senescence in MDA-MB-231 (Fig EV3B and C), whereas inactivation of let-7a reduces senescence and enhanced let-7a expression restricts proliferating MDA-MB-231 cells (Fig EV3D–F). Let-7a-induced increase in senescence could get further enhanced in MDA-MB-231 depleted for HuR (Fig EV3G). Thus, it is evident from these results that cell proliferation and senescence are regulated by HuR-driven export of let-7a in human breast cancer cell MDA-MB-231.

Does HuR regulate extracellular export of miRNA *in vivo*? To score the effect of starvation on liver miRNA level, BALB/c mice were starved for amino acids for 12 h and both tissue and serum level of miR-122 were measured. Following a similar trend to what was documented in Huh7 cells, increase in circulating miR-122 in the blood serum with associated loss of miR-122 in liver was noted in mouse Starved for 12 h. CAT-1 mRNA level was also conspicuously increased in Starved mouse liver (Fig 4A and B). We have also documented an increase in hepatic HuR level in Starved animals (Fig 4C). Such increase in HuR level or decrease in tissue miRNA level was not documented in the lungs of the Starved animals (Fig EV4A and B).

To confirm that HuR is sufficient for export of miRNAs from animal liver, we ectopically expressed HA-HuR in mouse liver by injecting the HA-HuR encoding expression plasmid through tail vein. After 4 days of injection, we documented expression of HA-HuR in the liver of treated group of animals with concomitant decrease in tissue miR-122 content and increase in target mRNAs level. It was accompanied by increase in circulating miR-122 content in the blood of the treated animals (Fig 4D–H). Neither such increase in lung HuR level nor any decrease in lung miRNA content was noted in corresponding animals injected with HA-HuR expression plasmids (Fig EV4C and D). Therefore, either starvation-induced increase or ectopic expression of HA-HuR can induce export of miR-122 from mouse liver.

## miRNA binding and replacement of Ago2 from target messages by HuR

Under stress condition, HuR binding with CAT-1 and RL-catA (reporter with 3′UTR CAT-1 mRNA) mRNAs were found to be increased. Having miR-122-binding sites in their 3′UTRs, these mRNAs were associated with Ago2 in non-stressed cells but showed reduced binding to Ago2 in stressed cells. RL-Con which does not have miR-122- or HuR-binding sites did not show detectable binding either to HuR or to Ago2 in Fed or Starved cells. Thus, HuR and miRNPs binding to common target RNAs was found to be mutually exclusive in Huh7 cells (Fig EV5A and B). HuR and Ago2 also do not show any interaction among themselves in either Fed or Starved hepatic cells [29,36]. We have tested the effect of HA-HuR expression on the binding of Ago2 to two different mRNAs with miR-122 sites. HA-HuR expression had no significant effect on Ago2 association of aldolase mRNAs. Aldolase mRNA does not have HuR-binding sites in their UTRs and is not a HuR-regulated gene. CAT-1 mRNA has both miR-122- and HuR-binding sites [29] and was less retained by Ago2 in cells expressing HA-HuR (Fig EV5C). Therefore, HuR could replace miRNPs from the co-targeted mRNA either in Starved or in HA-HuR overexpressing Huh7 cells.

We had detected HuR binding of miR-122 in Starved Huh7 cells (Fig 2A). Does HuR bind miRNAs by replacing them from Agos? To test, we have used isolated Ago2 miRNPs from HEK293 cells expressing miR-122 and incubated it with recombinant HuR in an *in vitro* reaction. We documented a time-dependent increase in HuR-associated miRNA with no detectable HuR-Ago2 interaction (Fig 5A). With a truncated version of HuR without the RRMIII domain, we detected almost 14-fold reduction in the capacity of truncated HuR to bind to the Ago2 replaced miR-122 compared to miRNAs bound by the full-length HuR. This argues in favor of the idea that the RRMIII of HuR plays an important role in HuR-miRNA bindings (Fig 5B–D). HuR could bind the miRNA either through a direct miRNA–HuR interaction or via an indirect binding facilitated through common target mRNAs. To resolute this issue, we performed RNA EMSA with $^{32}$P-labeled synthetic miR-122 and recombinant HuR or its truncated version and found a concentration-dependent increase in miR-122 binding of HuR (Fig 5E). The truncated version of HuR without the RRMIII showed a reduced miR-122 binding that is consistent with defective Ago2-miRNA displacement activity (Fig 5B–E). Both HuR and its truncated version, however, retain their capacity to bind the AU-rich element containing TNF-alpha RNA (Fig EV5E). Additionally, the binding of HuR to miR-122 could be partially inhibited by competitor TNF-alpha ARE RNA (Fig 5E).

Interestingly, the *in vivo* data suggest that all miRNAs possibly do not bind HuR with equal affinities as they all are not selected for export in Starved cells. We have done an *in vitro* binding assay of recombinant HuR with the small RNA pool of Huh7 cells isolated by miRVana miRNA isolation kit, and the HuR-bound RNAs were separated by immunoprecipitation using HuR-specific antibody. After amplification, relative $C_t$ values for different HuR-bound miRNAs were not proportional to their contents in the input RNA. As expected, miR-21 and miR-122 showed relatively higher abundance while miR-24 was not detected in HuR-associated RNA pool. The amount of HuR recovered U6 RNA was found to be very less and was in the non-reliable detection limit (Fig 5F).

Additionally, HuR-mediated uncoupling of miRNAs from Ago2 seems to be an irreversible process as excess expression of Ago2 could not restore the miR-122 level in Starved Huh7 cells. However, excess expression of Ago2 in HA-HuR-expressing cells reduces both the cellular and EV-associated miR-122 levels (Fig EV5D).

## Ubiquitination of HuR on MVB causes miRNA unbinding and export

Above experiments support HuR-mediated replacement of miRNAs from target mRNAs which facilitate enhanced extracellular export of miRNAs in Huh7 cells. OptiPrep™ gradient analysis that can separate subcellular organelles on a density gradient based on their densities [37] revealed relative increase in miR-122 content in late endosome/MVB-enriched fractions in HA-HuR-expressing cells, whereas Ago2 is largely endosomal (Figs 6A and C). This is consistent with increased miRNA export in HA-HuR-expressing cells as MVBs are the sites where miRNAs must accumulate before they get exported out. Interestingly, the HuR found on MVB fractions showed reduced binding for miRNA and majority of HuR in these fractions are of high molecular weight (Fig 6A and F). This suggests some possible post-translational modification of HuR happening in late endosome/MVB compartments. According to previous reports,

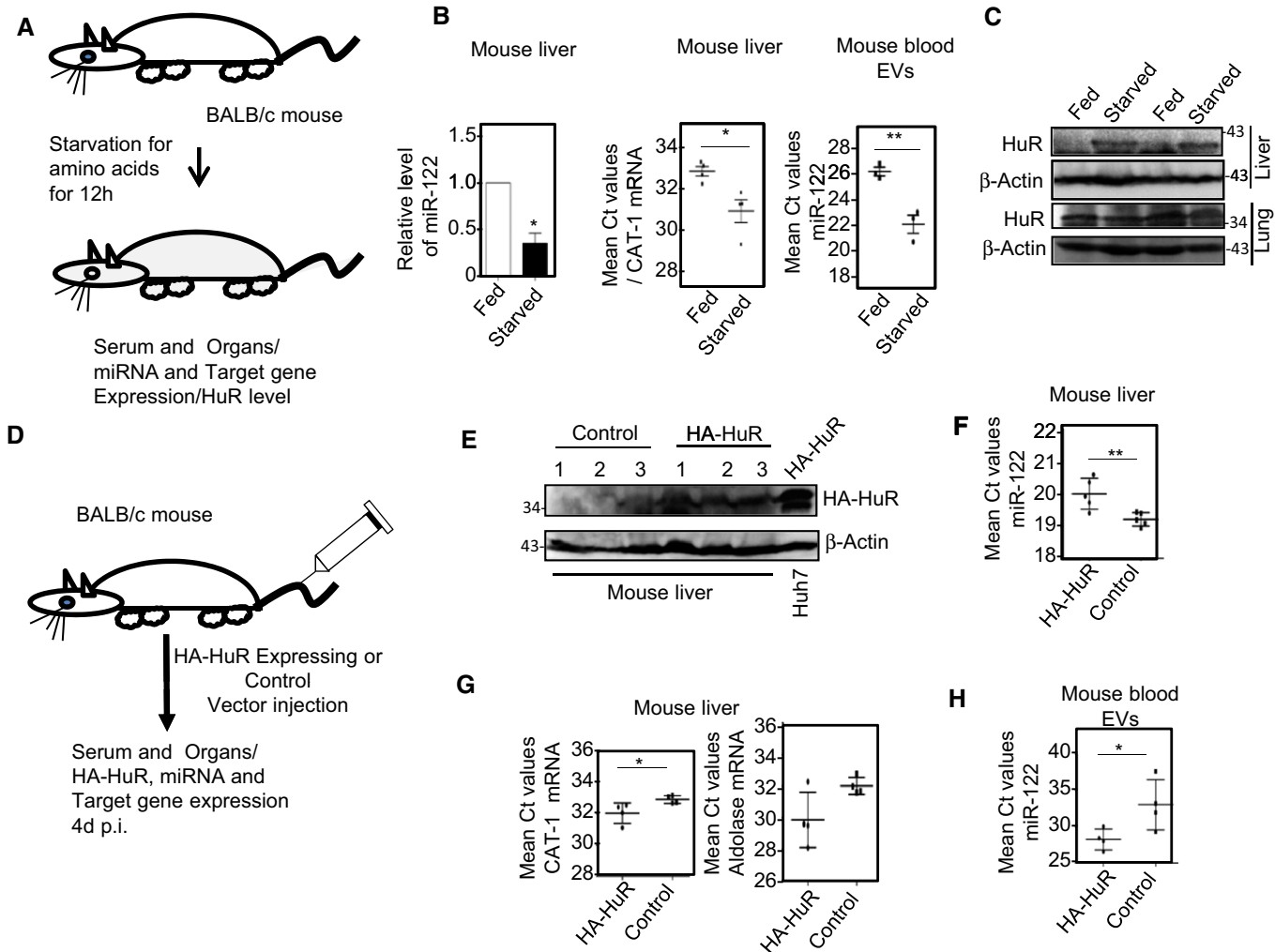

**Figure 4. HuR-driven miR-122 export from mouse liver enhances serum miR-122 content.**

A Schematic representation of starvation experiment in BALB/c mice.

B Change in miR-122 level in Starved mouse liver. Endogenous miR-122 (left panel) and CAT-1 mRNA (middle panel) levels in the liver of mouse Starved for 12 h were quantified by qRT–PCR. miR-122 levels in the blood serum of corresponding animals were also measured (right panel). The $C_t$ values of miR-122 and CAT-1 mRNA obtained for each animal liver and blood serum were plotted (mean ± s.e.m., *n* = 5).

C Relative levels of HuR in both liver and lung of Fed and Starved mice were analyzed by Western blots. β-Actin was used as loading control.

D Schematic representation of HA-HuR expression or control plasmid injection experiment in BALB/c mice.

E Western blot analysis to check expression of HA-HuR in liver samples of injected mice. β-Actin was used as loading control. HA-HuR-expressing Huh7 cell lysate was used as positive control.

F Change in miR-122 level upon expression of HA-HuR in mouse liver. Endogenous miR-122 level in HA-HuR expression plasmid-injected mouse liver was quantified by qRT–PCR and compared against liver RNA samples collected from control (pCIneo injected) group of animals (mean ± s.e.m., *n* = 5).

G Endogenous miR-122 target CAT-1 (left panel) and aldolase (right panel) mRNA levels in HA-HuR- or pCIneo (control)-injected mouse liver were quantified by qRT–PCR (mean ± s.e.m., *n* = 4).

H miR-122 levels in the blood serum of pCIneo (control) or HA-HuR expression plasmid-injected mice were measured. The $C_t$ values of miR-122 and CAT-1 and aldolase mRNAs obtained for each individual animal liver and blood serum were plotted (mean ± s.e.m., *n* = 5).

Data information: ns: non-significant, *$P$ < 0.05, **$P$ < 0.01. $P$-values were determined by paired *t*-test. d.p.i.: days post-injection. Positions of size markers in protein gels used for respective Western blot analysis are shown against each panel. For estimations of relative level of either miRNAs or mRNAs, $C_t$ values only within the range of 20–32 were considered for analysis.

ubiquitination of HuR plays important role in stress response regulation in mammalian cells [38–40]. To test whether there is any ubiquitination of HuR in Huh7 cells, we have expressed HA–ubiquitin and have documented ubiquitination of HuR (Fig 6B). To confirm the ubiquitination of HuR in Huh7 cells, we have performed *in vitro* ubiquitination assay of HuR by incubating the recombinant purified

HuR with the concentrated cell lysate of HA-Ub-expressing Huh7 cells in presence of ATP. After the *in vitro* reaction, recombinant HuR was immunoprecipitated with anti-His antibody and ubiquitinated HuR bands were detected. High molecular weight bands which were detected with anti-His antibody were also detected by HA-specific antibody which thereby confirmed the conjugation of

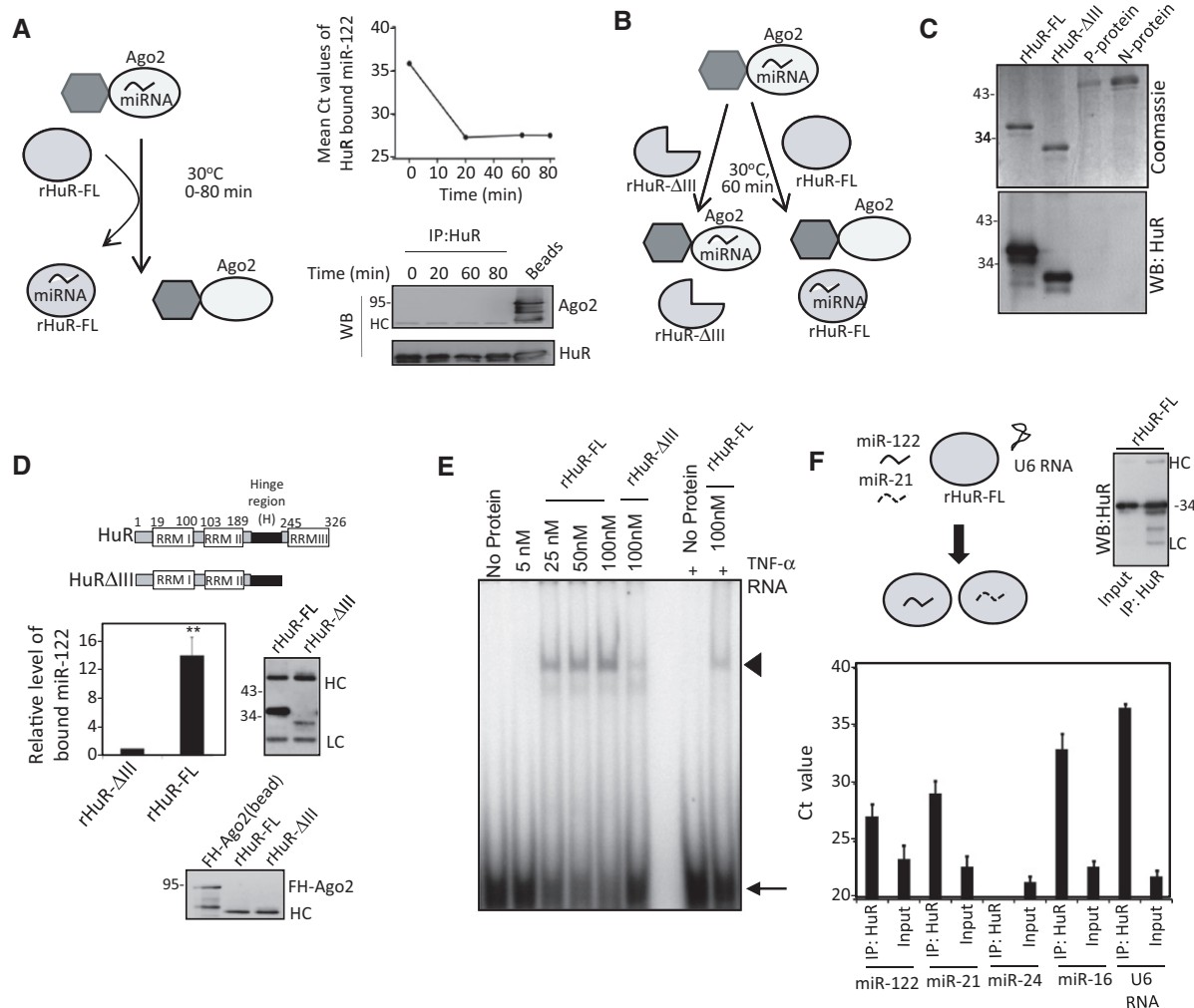

**Figure 5. HuR binds and replaces miRNA bound to Ago2.**

A   Time-course experiment of HuR binding of miR-122 replaced from Ago2 miRNPs. A schematic representation of *in vitro* miRNA-binding assay of HuR (left panel). Equal amounts of recombinant HuR (rHuR), after the indicated time of the miRNP interaction on FLAG beads, were immunoprecipitated, and HuR-associated miR-122 levels were detected by qRT–PCR (right top panel). Ago2 Western blot detects its presence in the FLAG beads used in the assay and its absence in HuR-immunoprecipitated materials. HuR Western blot was used to confirm its presence and quantification of its levels in the immunoprecipitated materials, at different time points (right bottom panel).

B   A schematic diagram of *in vitro* miRNA-binding assay of HuR, using wild-type full-length and delta RRMIII deletion mutant of HuR. Relative amount of miR-122 bound to full-length and delta RRMIII after 60 min of incubation with miR-122 containing miRNPs.

C   Coomassie-stained gel picture denotes the purity of the recombinant HuR used in the assay. The same proteins were Western blotted for HuR to show the specificity of the antibody used for immunoprecipitation. The recombinant P and N proteins of Chandipura virus were used as negative control to confirm the specificity of 3A2 anti-HuR antibody.

D   HuR Western blot in middle right panel confirmed the presence of HuR and its truncated version in the immunoprecipitated materials obtained from assays described in panel (B). A representative domain picture of HuR and HuRΔIII is shown in the upper panel. Relative levels of HuR-bound miR-122 were determined by qRT–PCR (mean ± s.e.m., *n* = 3) (middle left panel). Amount of miR-122 recovered from the immunoprecipitated materials was normalized against the amount of HuR or HuRΔIII present in the immunoprecipitated materials. Ago2 Western blot confirmed its presence in the FLAG beads used in the assay, while no Ago2 was detected in HuR-immunoprecipitated materials (bottom panel).

E   RNA electrophoretic mobility shift assay done with $^{32}$P-labeled synthetic miR-122 and recombinant HuR proteins. Radiolabeled miR-122 (100 fmol) was incubated with increasing concentration of FL-HuR (5–100 nM) or HuRΔIII (100 nM), and gel shifting of miR-122 was marked by arrowhead. Position of the free probes is marked by an arrow. In the right panel, 1 pmol of unlabeled TNF-α AU-rich element encoding synthetic RNA was used to compete with miR-122 binding of full-length HuR. Amount of HuR or HuRΔIII used for gel shift assay was premeasured by densitometric estimation of Coomassie-stained gel bands from SDS–PAGE.

F   Association of miRNAs with recombinant HuR *in vitro*. Small RNA pool of Huh7 cells isolated by mirVANA miRNA isolation kit was incubated with recombinant HuR in an *in vitro* binding reaction and HuR-associated RNA, recovered from immunoprecipitated HuR after the binding reaction, were quantified along with the input RNA used for binding. The mean $C_t$ values of RNA from samples from three independent binding reactions along with input RNA were measured and plotted (mean ± s.e.m., *n* = 3). Recombinant HuR used in the binding assay and in the immunoprecipitated materials was detected by Western blot in the input.

Data information: **$P < 0.01$. *P*-values were determined by paired *t*-test. Positions of size markers in protein gels used for respective Western blot analysis are shown against each panel. HC: heavy chain and LC: light chain of IgG used for immunoprecipitation reaction. For estimations of relative level of miRNAs, $C_t$ values only within the range of 20–32 were considered for analysis.

Source data are available online for this figure.

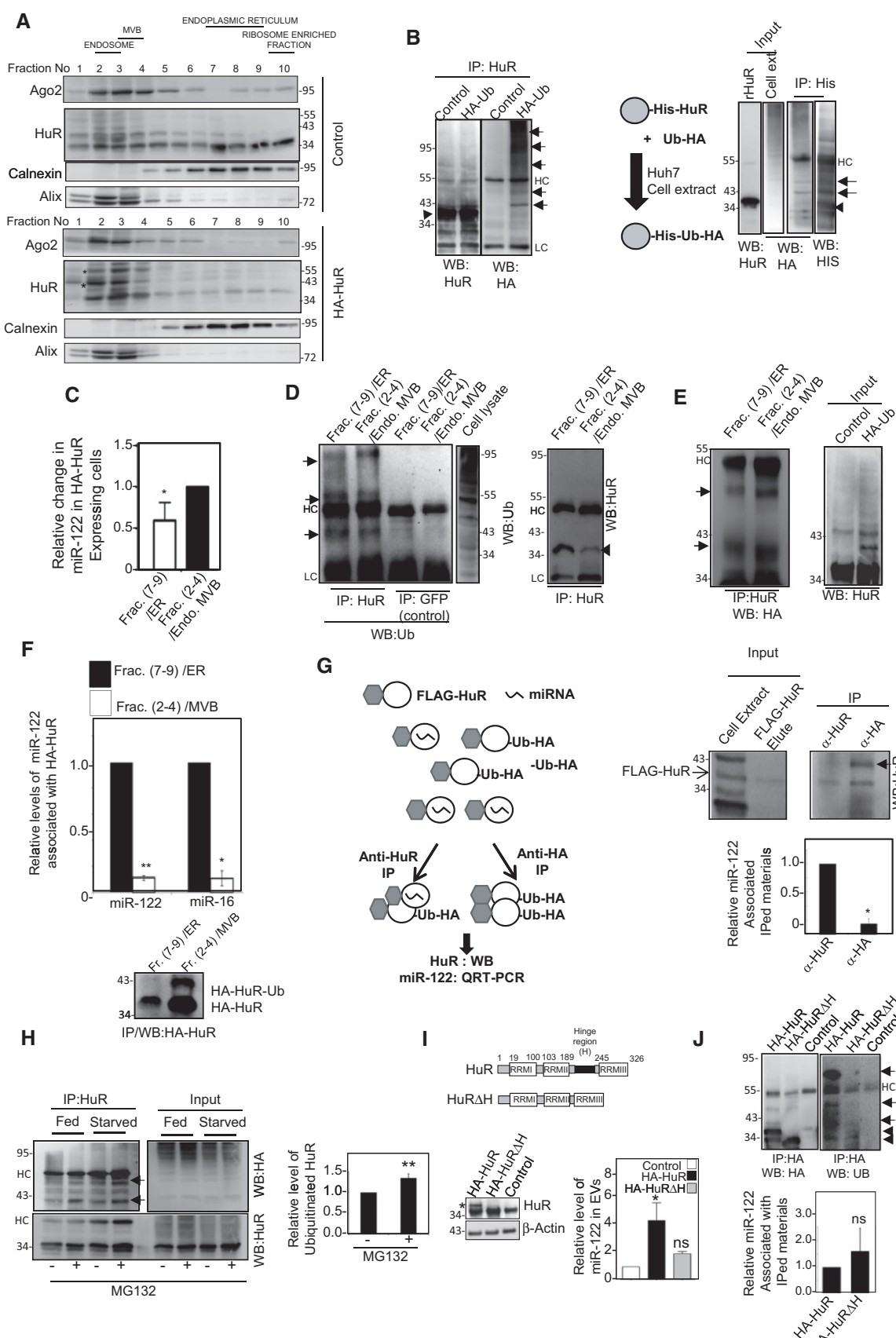

Figure 6.

**Figure 6.  Ubiquitination-dependent miRNA unloading of HuR on MVB.**

A   Subcellular distribution of HuR and Ago2 in Huh7 cells. Isotonic lysates of Huh7 cells transfected with HA-HuR expression or pCIneo control vector were analyzed on 3–30% OptiPrep™ gradients for separation of organelles and localization of Ago2 and HuR were determined by Western blotting analysis. Alix and calnexin were used as markers of late endosome/MVB and ER, respectively. Positions of specific organelle containing fractions are marked above the panels. * denotes the position of the high molecular weight HuR bands detected in the MVB fractions.

B   Detection of ubiquitinated HuR in Huh7 cells expressing HA-Ub. HuR was immunoprecipitated with anti-HuR antibody from cell lysate of Huh7 cells transfected with HA-Ub expression or pCIneo (control) plasmids and Western blotted for HA or HuR (left panel*). In vitro* ubiquitination assay was done with recombinant His-tagged-HuR and cell extract of Huh7 cells expressing HA-Ub. The scheme of the experiment and Western blot data for input and cell extract along with immunoprecipitated materials are shown (middle and right panel). Specific bands of Ub-HuR are marked by arrows and HuR is marked by arrowhead.

C   Changes in relative level of miR-122 in the subcellular ER- and MVB-associated fractions of OptiPrep™ gradients done with the cell lysates from HA-HuR expression vector-transfected cells were measured by qRT–PCR (mean ± s.e.m., n = 5).

D   Detection of ubiquitinated HuR in HuR-immunoprecipitated materials obtained from ER- and MVB-enriched fractions of Huh7 cells. Organelles were separated on OptiPrep™ gradients, and HuR in organellar lysates was immunoprecipitated with HuR-specific antibody and Western blotted for ubiquitin to detect ubiquitinated form of HuR. Whole-cell lysates used for OptiPrep™ analysis were Western blotted for ubiquitinated proteins. Positions of the high molecular weight bands of ubiquitinated HuR are marked by arrows. Levels of HuR in the immunoprecipitated materials were detected using anti-HuR antibody and marked by arrowheads. Anti-GFP antibody-immunoprecipitated materials were used as controls.

E   Detection of ubiquitinated HuR in HuR-immunoprecipitated materials obtained from ER- and MVB-enriched fractions of cells co-expressing HA–ubiquitin and FLAG-HuR. Cells were lysed and organelles were separated on OptiPrep™ gradients, and HuR in organellar lysates was immunoprecipitated with HuR-specific antibody and Western blotted for HA to detect ubiquitinated form of HuR. HuR in the lysate of the FLAG-HuR and HA-Ub co-expressing cells was detected by HuR-specific antibody. Positions of the Ub-HuR bands are marked by arrows.

F   Loss of miRNA association of HuR on MVB. HA-HuR-expressing Huh7 cell lysates separated on OptiPrep™ gradients were used for immunoprecipitation of HA-HuR and associated miRNA levels were measured by qRT–PCR and normalized against immunoprecipitated proteins. Levels of immunoprecipitated HA-HuR were detected by Western blotting and quantified by densitometry. Data are represented as mean ± s.e.m., n = 3.

G   Loss of miRNA from ubiquitinated form of HuR. Scheme of the double-immunoprecipitation experiment is shown in the left panel. Western blot analysis for HuR in the anti-HA and anti-HuR-immunoprecipitated FLAG-HuR along with the input of FLAG-HuR used. Lysates of cells expressing FLAG-HuR and used for FLAG-HuR purification are also shown (right upper panels). qRT–PCR-based estimation of the miRNA per unit of HuR-immunoprecipitated for both HA and HuR-immunoprecipitated materials and normalization against the respective protein levels quantified from Western blots (right lower panel). Data are represented as mean ± s.e.m., n = 3. Position of ubiquitination band of FLAG-HuR is marked by arrow. The cell lysate expressing the HA-Ub and FLAG-HuR along with purified FLAG-HuR used in immunoprecipitation experiment are shown as input in left upper panel. Position of the FLAG-HuR is marked by an open arrow.

H   Increase in ubiquitinated form of HuR in Starved Huh7 cells. Levels of ubiquitinated HuR in Fed and Starved (5 h) HA-Ub-expressing Huh7 cells treated either with ethanol or with MG132 (20 μM) for the same duration. The lysates were immunoprecipitated with HuR and Western blotted for HA and HuR. Positions of ubiquitinated bands are marked by arrows. The levels of ubiquitinated HuR were measured by densitometry and relative increase in ubiquitinated HuR levels against the non-ubiquitinated form was measured and plotted (mean ± s.e.m., n = 3).

I   Full-length and a truncated version of HA-HuR without the hinge region (graphical scheme in upper panel) was used to score the effect of deletion of hinge region on EV-associated miR-122 content measured by qRT–PCR (bottom panel). Western blots to check expression of HuR and its truncated variant are also shown. * denotes the HA-HuR band (left lower panel). Data are represented as mean ± s.e.m., n = 3. β-Actin served as loading control.

J   Ubiquitination status of full-length and a truncated version of HA-HuR without the hinge region between RRMII and RRMIII in Huh7 cells. Full-length or truncated version of HuR was immunoprecipitated from cell lysates of Huh7 cells expressing these proteins, and the immunoprecipitated materials were Western blotted for HA-HuR and ubiquitin. Ubiquitinated HuR bands are marked by arrows. Positions of non-ubiquitinated HuR proteins are marked by arrowheads (upper panel). qRT–PCR-based estimation of associated miRNA per unit of immunoprecipitated materials was estimated and normalized against the respective protein levels quantified by densitometric quantification of Western blots (lower panel). Data are represented as mean ± s.e.m., n = 3. PCIneo-transfected cells were used as control in immunoprecipitation reaction, but no amplification of miR-122 was detected for HA immunoprecipitated materials from pCIneo-transfected cells.

Data information: ns: non-significant, *P < 0.05, **P < 0.01. P-values were determined by paired *t*-test. Positions of size markers in protein gels used for respective Western blot analysis are shown against each panel. HC: heavy chain and LC: light chain of IgG used for immunoprecipitation reaction. For estimations of relative level of miRNAs, $C_t$ values only within the range of 20–32 were considered for analysis.

Source data are available online for this figure.

---

HA-Ub with His-HuR *in vitro*. These bands were originally absent both in the cell extract or in recombinant HuR used in the assay (Fig 6B).

Interestingly, ubiquitinated HuR was predominantly associated with MVB/late endosome fractions and MVB-associated HuR has shown reduced binding to miRNAs (Fig 6D–F). In Huh7 cells, not overexpressing HuR, relative low amount of HuR was found to be associated with the MVB fractions but the MVB-associated HuR is predominantly in ubiquitinated form (Fig 6A and D). From the above data, it seems that ubiquitination may uncouple miRNA from HuR. To confirm reduced miRNA binding by ubiquitinated form of HuR, we co-expressed HA–ubiquitin and FLAG-HuR in Huh7 cells. Using affinity-purified FLAG-HuR isolated from Huh7 cells, we immunoprecipitated HuR either with an anti-HuR or with an anti-HA-specific antibody. Anti-HA antibody should immunoprecipitate only the ubiquitinated form of HuR. Estimation of the miRNA content of the immunoprecipitated materials showed reduced

miRNA binding of FLAG-HuR that was immunoprecipitated with anti-HA antibody (Fig 6G). Increased ubiquitination of HuR was also documented in Starved Huh7 cells compared to Fed (control), where ubiquitinated HuR increases in presence of proteasomal inhibitor MG132 that prevents degradation of ubiquitinated HuR (Fig 6H).

Hinge region of HuR is required for replacement of miRNPs from the target messages [36]. Importantly, the 110 amino acids long segment shown to be required for ubiquitination of HuR also span to the hinge region [38]. HA-HuRΔH, the deletion mutant of HuR without the hinge region between RRMII and RRMIII domains, was found to be less effective in promoting extracellular export of miRNA (Fig 6I). Consistent with this observation, reduced level of ubiquitination was documented for HA-HuRΔH but the miR-122-binding capacity of this mutant remained largely unaffected (Fig 6J). Therefore, defective ubiquitination of this deletion mutant of HuR may account for reduced extracellular export of miR-122 in

Huh7 cells expressing the hinge region truncated version of HuR (Fig 7).

## Discussion

This manuscript describes how the cellular miRNA levels respond to cellular needs and how a stress responsive protein HuR augments the export process of a miRNA to buffer its cellular level. This work connects the previously reported HuR-mediated derepression of miRNA–target messages [29] to the export of corresponding miRNAs that are replaced by HuR from targeted messages.

Under stress condition, rapid post-transcriptional changes in mRNA stability and translation are the primary mechanisms that ensure prompt response. This occurs by reorganizing the translational machineries and translatable mRNA pool under changed cellular environment. Extracellular export of repressive miRNAs with enhanced expression of their targets under stress condition could ensure simultaneous upregulation of several genes that are repressed by that same miRNA in non-stressed cells. HuR, being a key protein component in replacing the miRNPs from its target messages, is expected to play a pivotal role in export of miRNAs that are bound to messages co-regulated by HuR. However, it would be interesting to identify the basis of HuR specificity for miRNAs binding and export. Our data suggest that HuR unloads the miRNPs from co-targeted messages. It also binds replaced miRNAs to facilitate their export.

miRNA binding of HuR was reported earlier and lowering of cellular levels of several miRNAs that are identified as HuR targets were reported [31]. However, no mechanism was proposed for HuR-mediated miRNA reduction. How does HuR augment the uncoupling of miRNPs? It is possible that HuR may do so by shifting the equilibrium to a miRNA unbound state of Ago proteins by capturing the Ago-free miRNAs released from miRNA–Ago complexes.

As targeted degradation of miR-122 could be a major contributor in lowering of miRNA levels in Starved Huh7 cells, we also explored the possibility of enhanced miR-122 degradation in Starved hepatic cells. Unlike the expected enhanced rate of miR-122 degradation in Starved cells, we observed similar reductions both for miR-122 and for CAT-1 mRNA in GW4869-treated cells co-treated with α-amanitin (Fig EV1E). This suggests that the degradation rate of miR-122 is similar in Fed and Starved Huh7 cells pre-blocked for EV-mediated export of miR-122 by GW4869. Therefore, starvation-induced lowering of cellular miR-122 is primarily contributed by an increase in EV-associated miR-122. This is consistent with changes in the copy number of miR-122 in Starved Huh7 cells and its corresponding increase in the EVs. The cellular copy numbers of miR-122 drop from 12,000 to 5,000 upon starvation, while the increase in EV content could account for 50% of that cellular decrease. Considering the half-life of a miRNA in EVs is about twofold less than its cellular half-life (K. Mukherjee et al, unpublished results), the EV-associated miR-122 increase can account for almost the entire amount of miR-122 that gets reduced in Starved Huh7 cells. Interestingly, although cellular

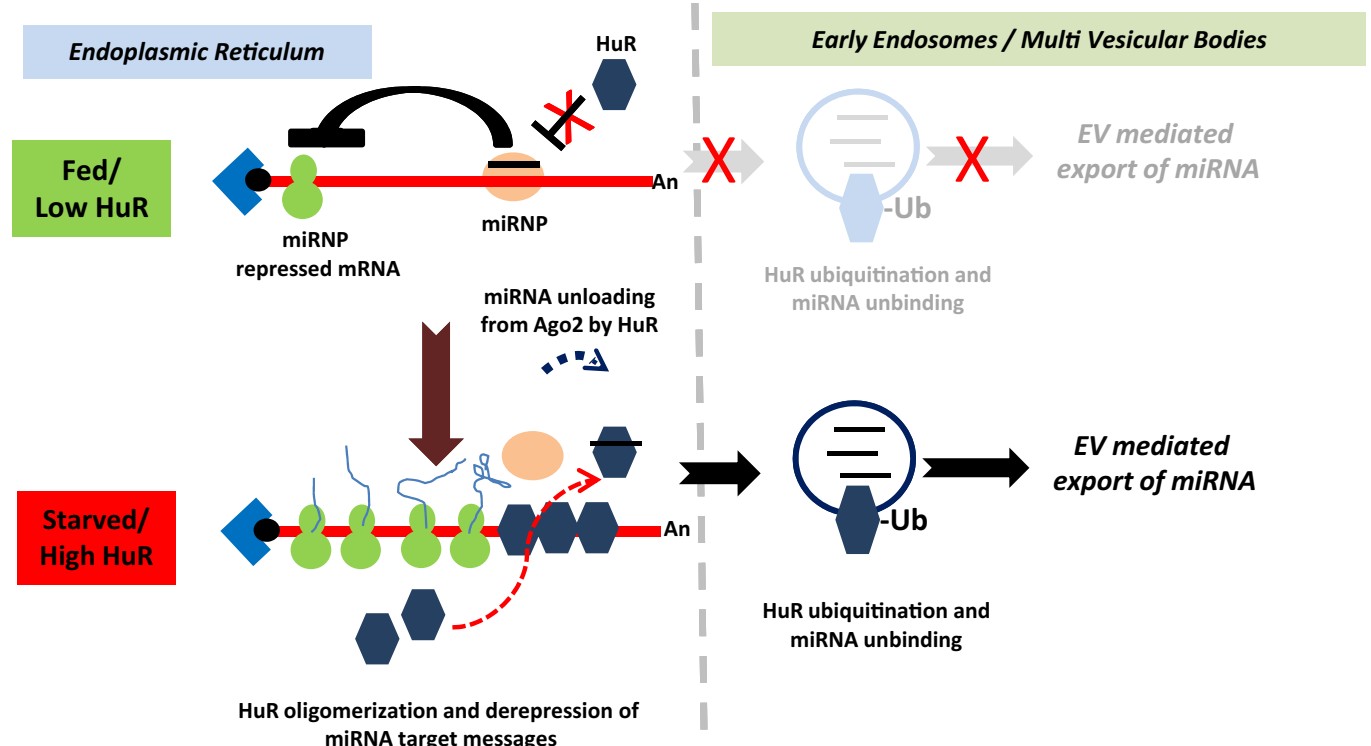

**Figure 7.    A possible model of HuR-driven extracellular export of miRNA in human cells.**
Under Fed condition, miRNPs remain bound to target messages and show reduced extracellular export. In cells Starved for amino acids and metabolites, HuR binding to co-targeted messages replaces miRNPs from target messages. HuR binds miRNAs and replaces miRNAs from miRNPs. In subsequent step, ubiquitination of HuR uncouples bound miRNAs on MVB and facilitates extracellular export of HuR-decoupled miRNAs.

CAT-1 and aldolase mRNA levels increase with starvation, we could not detect any CAT-1 or aldolase mRNAs in the EVs isolated from Fed or Starved Huh7 cells. Levels of GAPDH mRNA in EVs do not change upon starvation (K. Mukherjee et al, unpublished results).

Factors identified as regulators of extracellular export of miRNA in animal cells are limited. In a recent report, role of HuR as "miRNA sponge" for regulation of miRNA activity has been documented, where HuR binds specific miRNA to reduce its effect on its targets [32]. In other context, ubiquitination of HuR has been reported to have a modulatory role in HuR function during stress [38,39]. Correlating these observations, we have documented increased miR-122 binding of HuR in stressed cells happening primarily on ER, while ubiquitinated HuR is associated with MVBs. Ubiquitination ensures uncoupling of HuR-bound miRNAs which in turn leads to enhanced export of these HuR-decoupled miRNAs. Therefore, identification of the role of HuR in extracellular export of miRNAs paves the path for future research on other such factors that may affect EV-mediated export process by targeting miRNA binding of HuR. Previously, modifications as well as trimming and tailing of miRNAs have been reported that could make them either degradable or exportable in mammalian cells [41]. Through high-resolution Northern blot analysis, we could not detect such a change for miR-122 either in cellular or in exosomal pools in Fed or Starved condition (Fig 1A). Therefore, the post-transcriptional modifications of miR-122 are not the signals that are recognized by HuR to select them for export. However, miRNA binding of HuR and its exosomal export may not always be directly coupled as additional factors may affect the export path. Downregulation of certain miRNAs like miR-21 in EVs upon HuR depletion does not always cause an increase in the cellular pool of respective miRNAs. Existence of additional mechanism such as transcriptional level feedback loop may operate to adjust the cellular miRNA levels for those miRNAs.

Role of Ago2 in HuR-driven export of miRNAs is also not clear. In a cellular context where Ago2 is expressed in excess, the miRNA export process seems to be independent of cellular Ago2 content both in stressed or in HA-HuR-expressing cells. This is consistent with the idea that HuR-driven miRNA unloading from Ago2 complex is an irreversible process. This may be due to differential compartmentalization of HuR and uncoupled Ago2 in Huh7 cells. Thus, miRNAs, uncoupled from Ago2, that are once taken to endosome/MVB compartment by HuR cannot get reloaded to Ago2. The mechanistic details of such regulation will be of interest for further exploration.

The $K_d$ for miR-122 binding to HuR is found to be 50 nM, that is comparable to $K_d$ values of miRNA binding to Ago2 (20–80 nM as shown in [42] and [43]). Therefore, HuR should be able to compete with Ago2 for miRNA binding. Additionally, miRNA/HuR-binding $K_d$ is in the same range to what has been observed previously for other canonical HuR-binding element containing RNAs [36]. Evidences for HuR specificity for its substrates are substantial, and numerous papers have already been published which confirm that HuR bindings with its targets are specific. In this context, Poria et al [32] has already highlighted the specificity of HuR-miRNA binding through EMSA assay. Selectivity of HuR for its target miRNAs also get supported by the other experiments that we have done in this manuscript when we measured HuR-bound miRNA levels that could vary in different context (immunoprecipitation experiments in Fed and Starved cells, Fig 2A) or in vitro (Figs 5F and EV5F). The binding of HuR to the miRNAs is mediated by its

RRMIII domain as mutant of HuR devoid of the RRM III could not bind miRNA.

HuR preferentially binds to the target mRNA having AU-rich sequences. There are also evidences that HuR could bind the GU-rich sequence (GRE) on a RNA where GRE binding by the protein itself controls alternative splicing of HuR mRNA in mammalian cells to regulate its own expression [44]. We found GU richness both in miR-122 and in miR-21 that could explain their preferential association with HuR ([32] and Fig 5F). Therefore, it is possible that RRMIII of HuR, that is known to play a role in its interaction with poly A stretch, interacts with miRNAs with GU-rich sequence and selects them for export.

EMSA assay done in presence of miR-122 antisense RNA (miR-122*) depicts complete inhibition of miR-122 binding of HuR when substrate RNA is present in excess. Therefore, the HuR binding to miRNA should also get affected by the abundance of its substrates. Interestingly, the miR-122 antisense RNA too contains a stretch of GU-rich sequence, therefore also showed weak binding to HuR ($K_d$ = 200 nM). As HuR specifically binds to single-stranded RNA and can also bind the miR-122* RNA, HuR may play a role in controlling degradation or export of miRNA* strands and thus could regulate miRNA biogenesis steps in animal cells.

From our results, it seems that HuR gets targeted to late endosomes and MVBs for ubiquitination and subsequent miRNA unbinding. This is an additional layer of gene regulation where specific factors contribute and ensure targeting and ubiquitination of HuR on MVBs. Identification of specific factors and ubiquitin ligase responsible for ubiquitination of HuR will be useful in tracking the entire pathway of miRNA export. It would be interesting to identify whether the miRNA bound form of HuR gets preferentially targeted or sorted to MVBs for ubiquitination compared to the miRNA-free form of HuR. It may also be important to note that in ER fraction, some of the associated HuR are ubiquitinated. That may be due to the shuttling of ubiquitinated and miRNA unbound form of HuR to ER before their deubiquitination.

ESCRT proteins have been previously documented for their role in selecting the protein cargos for MVBs, and in this process, ubiquitination of proteins acts as a signal for MVB targeting and also for exosomal export or lysosomal degradation [45]. Here, we report ubiquitination of HuR serves as a novel mechanism for selective loading of HuR-associated miRNAs to EVs. Identification of the factors including the ESCRT components that may regulate the ubiquitination and miRNA unloading of HuR will be important to identify.

miRNA-loaded EVs are used for intercellular communications where the retaken EV content is reused in recipient cells [25,46]. The extracellular release of miRNAs may ensure the homogeneity of miRNA expression and activities among neighboring cells. In a tissue, the spiked expression of target mRNAs in individual cells of a population may be tamed by the extra miRNAs received via EVs till the balanced expression of target genes and miRNAs is achieved. From the presented data, it is also clear that EV-mediated export of miRNA let-7a controls senescence and proliferation in human breast cancer cell MDA-MB-231. HuR also plays a role in controlling let-7a export to control senescence in this cancer cell.

HuR and miR-122 both have a positive influence on HCV replicon RNA levels in Huh7 cells [47,48]. We tested the effect of HCV replicon expression on cellular miR-122 level co-expressing Myc-HuR and have noted a similar decrease in miR-122 content as

documented in non-HCV-infected Huh7 cells (Figs 3C and EV3H–J). Therefore, it is likely that HCV-HuR interaction may happen at different subcellular location that precludes its effect on HuR-miR-122 interaction—necessary for miR-122 export.

In spite of some recent findings, the mechanism of sorting of miRNAs to exosomes is not entirely clear [49,50]. The importance of Rab27 proteins in controlling the exosome sorting has been explored, but their role in miRNA targeting is still obscure [51]. It is also not entirely clear how specific miRNAs get targeted for export. Target availability has been reported to have an influence on miRNA stability in neuronal cells [52]. How the availability of target messages will affect the EV targeting of miRNAs is also an interesting but less explored question. Only recently, Squadrito *et al* [53] have reported how availability of the target messages affects the sorting of miRNAs to exosomes and its delivery to target cells. This interesting aspect has not been explored in this manuscript in an *in vivo* context. However, from the *in vitro* binding assays, it seems that availability of target RNAs that could bind the miRNAs to form strong hybrids can inhibit HuR-miRNA binding. This is consistent with the hypothesis that target RNA may negatively influence the HuR-driven export of miRNAs in mammalian cells.

Heterogeneity of exosomes carrying different cargo proteins is of interest as it is not clear whether all the exosomes present in a population are functionally distinct although they all may be positive for a specific marker protein.

Here, we focused on how the extracellular export of miRNAs is controlled in animal cells. To that end, we have identified HuR, a well-explored RNA-binding protein known to stabilize ARE-containing messages in mammalian cells, as a regulator of extracellular export of miRNAs. Apart from providing evidence for the new role of HuR as a controller of miRNA stability, this work also provides a possible molecular link between contextual binding of a miRNA to its target and its export. These findings could explain why export and hence turnover of miRNAs are selective in animal cells. In this work, we have documented EV-mediated miRNA export as a mechanism used by human cells to regulate its cellular miRNA content. We also have delineated the molecular mechanism related to this phenomenon happening in human hepatic cells to augment stress response under starvation (Fig 7).

## Materials and Methods

### Expression plasmids, cell culture, starvation, and transfection

Information of all plasmids and siRNAs used in this study is available in the Table EV1. Information of all DNA oligos used as primers for real-time quantification and gene amplification is listed in Table EV2. All human cell lines were grown in Dulbecco's modified Eagle's medium (DMEM) supplemented with 2 mM L-glutamine and 10% heat-inactivated fetal calf serum (FCS). For starvation experiments, cells were incubated in Hank's balanced salt solution (HBSS) supplemented with 10% dialyzed FCS (all from Invitrogen). Exosomal inhibitor GW4869 was used at a final concentration of 10 μM (Calbiochem). Thapsigargin (Calbiochem) was applied at 2.5 μM for 12 h, and MG132 was used at 20 μM concentration for 5 h. For all experiments, cells were grown to 25–60% confluent states unless specified otherwise.

Transfections of HEK293, MDA-MB-231, and Huh7 cells all were performed using Lipofectamine 2000 (Invitrogen) according to manufacturer's protocol. For target RNA or HA-HuR overexpression experiments, 2 μg of respective plasmids was transfected per well of a six-well plate in triplicates. For immunoprecipitation, FH-Ago2 plasmids were either co-transfected with pCIneo control vector or with HA-HuR expression plasmid. siRNAs were transfected at 50 nM concentration using either Lipofectamine RNAiMAX or Lipofectamine 2000. AllStars Negative Control siRNA (Qiagen) was used as control (siCon). Huh7.5 monolayer cells used for HCV replication were maintained in DMEM (Sigma) with 10% fetal bovine serum (GIBCO, Invitrogen). They were transfected with *in vitro* transcribed HCV-JFH1 RNA using Lipofectamine 2000 reagent (Invitrogen) in antibiotic-free medium. Six hours post-transfection, Dulbecco's modified Eagle's medium supplemented with 10% fetal bovine serum was added. The cells were harvested using TRI reagent (Sigma) for total RNA isolation and qRT–PCR.

### Northern and Western blot

Northern blotting of total cellular RNA (5–15 μg) was performed as described previously [10,54]. For miRNA detection, $^{32}$P-labeled 22 nt antisense DNA- or LNA-modified probes specific for respective miRNAs or U6 snRNA were used. PhosphorImaging of the blots was performed in Cyclone Plus Storage Phosphor System (PerkinElmer), and OptiQuant software (PerkinElmer) was used for quantification.

Western analyses of different proteins were performed as described previously [10,54]. Detailed list of antibodies used for Western blot and immunoprecipitation is available in Table EV3. Imaging of all Western blots was performed using an UVP BioImager 600 system equipped with VisionWorks Life Science software (UVP) V6.80.

### EV preparations and treatments

Extracellular vesicles were isolated based on the published protocols [55]. Briefly, for isolation of EVs, cells were grown in medium pre-cleared for the same and the culture supernatants were clarified for cellular debris and other contaminants by centrifugation at 400 × *g* for 5 min, 2,000 × g for 10 min, and 10,000 × g for 30 min and filtration through a 0.22-μm filter. EVs were separated by centrifugation at 100,000 × *g* for 90 min or alternatively by precipitation using the System Biosciences Exo-Quick$^{TC}$ Exosome Precipitation Solution. EV pellets were resuspended in 200 μl of 1× passive lysis buffer (Promega) for isolation of proteins and RNA. Characterization of isolated EVs and its purity check were done as the procedures adopted elsewhere [25].

Extracellular vesicles from HA-HuR and pCIneo-expressing cells were purified by immunoprecipitation using biotinylated capture tetraspanin (CD63) antibody by System Biosciences Tetraspanin Exo-Flow capture Kit as per manufacturer's protocol. The EV-associated RNA was isolated using Trizol LS, and relative miRNA level was quantified. Affinity-purified extracellular vesicles (EVs) from Fed and Starved samples were characterized by FACS analysis. For characterization of EVs by floatation technique, culture supernatants from four 90-mm dishes culture of Huh7 cells (80% confluent) expressing either HA-HuR or pCIneo were collected and extracellular vesicles were isolated by ultracentrifugation method

and purified on a sucrose cushion (1 M sucrose and 10 mM Tris–HCl pH 7.5). OptiPrep™ (Sigma-Aldrich, USA) was used to prepare a 5–40% increasing discontinuous gradient in a buffer containing 250 mM sucrose and 10 mM Tris (pH 7.5) for separation of EVs as described previously with minor modifications [56,57].

The EV containing pellet was resuspended in a buffer containing 250 mM sucrose and 10 mM Tris–HCl (pH 7.5), and this solution was loaded on top of 5–40% increasing discontinuous gradient and was ultracentrifuged for separation of gradient at 120,000 × g for 7.5 h using established protocols and ten fractions were collected [56,57]. RNA was isolated from each fraction using Trizol LS and relative miRNA levels were quantified by qRT–PCR. Relative CD63 (EV-marker protein) level was analyzed by Western blotting using the anti-CD63 antibody.

## Fractionation and separation of multivesicular bodies, endosomes, and ER on OptiPrep™ gradients

OptiPrep™ (Sigma-Aldrich, USA) was used to prepare a 3–30% continuous gradient in a buffer containing 78 mM KCl, 4 mM MgCl$_2$, 8.4 mM CaCl$_2$,10 mM EGTA, and 50 mM HEPES-NaOH (pH 7.0) for separation of subcellular organelles as described previously with minor modifications stated below [10]. Cells were trypsinized, washed, and homogenized with a Dounce homogenizer in a buffer containing 0.25 M sucrose, 78 mM KCl, 4 mM MgCl$_2$, 8.4 mM CaCl$_2$, 10 mM EGTA, 50 mM HEPES-NaOH (pH 7.0) supplemented with 100 μg/ml of cycloheximide, 5 mM vanadyl ribonucleoside complex (VRC) (Sigma-Aldrich), 0.5 mM DTT, and 1× protease inhibitor cocktail (Roche). The lysate was clarified by centrifugation at 1,000 × g for 5 min and layered on top of the prepared gradient and was ultracentrifuged for separation of gradient using established protocols, and ten fractions were collected.

## Immunoprecipitation (IP) and analyses of associated mRNA and miRNA by qRT–PCR

For IP reactions, FLAG-HA-Ago2 (FH-Ago2)-expressing cells were lysed in lysis buffer [20 mM Tris–HCl, pH 7.5, 150 mM KCl, 5 mM MgCl$_2$, 2 mM DTT, and 40 U/ml RNase inhibitor, 1% Triton X-100, and 1× EDTA-free protease inhibitor cocktail (Roche)] for 30 min at 4°C. The lysates were clarified by sonication (three pulses of 10 s each) followed by centrifugation at 16,000 × g for 10 min at 4°C. Protein G Agarose beads (Invitrogen) were pre-blocked with 5% BSA in lysis buffer for 1 h followed by binding of specific antibody (1:50) for 4 h at 4°C. Subsequently, the pre-cleared lysates were added to the antibody bound beads and incubated overnight at 4°C on a rotator. Beads were washed thrice with IP buffer (20 mM Tris–HCl pH 7.5, 150 mM KCl, 5 mM MgCl$_2$, 2 mM DTT), and the bound proteins were analyzed by Western blot. In parallel, from half of the immunoprecipitated materials, separated during washing steps, RNA was extracted with Trizol LS (Invitrogen). mRNA and miRNA levels were quantified by two-step qRT–PCR. Real-time analyses by two-step RT–PCR were performed for quantification of miRNA and mRNA levels on a 7500 REAL TIME PCR SYSTEM (Applied Biosystems) or QuantStudio 12K Flex Real-Time PCR System or Bio-Rad CFX96™ real-time system using Applied Biosystems Taqman chemistry-based miRNA assay system. mRNA real-time quantification was generally performed in a two-step format using Eurogentec Reverse

Transcriptase Core Kit and MESA GREEN qPCR Master Mix Plus for SYBR Assay with Low Rox kit from Eurogentec following the suppliers' protocols. The comparative $C_t$ method which typically included normalization by the 18S or GAPDH RNA levels for each sample was used for relative quantification.

miRNA assays by real-time PCR were performed with 25 ng of cellular RNA and 200 ng of EV RNA unless specified otherwise, using specific primers for human let-7a (assay ID 000377), human miR-122 (assay ID 000445), human miR-16 (assay ID 000391), human miR-21 (assay ID 000397), and human miR-24 (assay ID 000402). U6 snRNA (assay ID 001973) was used as an endogenous control. One-third of the reverse transcription mix was subjected to PCR amplification with TaqMan® Universal PCR Master Mix No AmpErase (Applied Biosystems) and the respective TaqMan® reagents for target miRNA. Samples were analyzed in triplicates from minimum three biological replicates. The miRNA levels were defined from the cycle threshold values ($C_t$) for representation of exosomal miRNA or immunoprecipitated miRNA levels. The comparative $C_t$ method, which typically included normalization, by the U6 snRNA, or a non-relevant miRNA, for each sample was used for all other instances. HCV-negative strand RNA was quantified by two-step qRT–PCR using DyNAmo™ HS SYBR® Green qPCR Kit (Finnzymes). cDNA was synthesized with Moloney murine leukemia virus (M-MLV) reverse transcriptase at 42°C for 1 h (Promega) using 1 μg of total RNA by adding primers targeting HCV-negative strand and glyceraldehyde 3-phosphate dehydrogenase (GAPDH) mRNA in same reaction. Quantitative reverse transcription PCR (qRT–PCR) was done using the cDNA in a 10 μl reaction according to manufacturer's instructions for 40 cycles. Comparative threshold cycle ($C_t$) method was used to calculate fold change in HCV RNA level ($2^{(-\Delta\Delta Ct)}$) and normalized with GAPDH.

### *In vitro* miRNA displacement assay

FLAG-HA-Ago2-transfected HEK 293T stable cell line was transfected with pre-miR-122-expressing pmiR-122 plasmid (1 μg/well) in a six-well format using Lipofectamine 2000 following manufacturer's protocol. Cells were split after 24 h of transfection and were harvested and lysed after 48 h of transfection in the lysis buffer (20 mM Tris, pH 7.5, 5 mM MgCl$_2$, 150 mM KCl, 0.5% Triton X-100, 0.5% sodium deoxycholate, 1× PMSF, 2 mM DTT, 40 U/ml RNase inhibitor) for 30 min on a neutator at 4°C, followed by centrifugation at 3,000 × g for 10 min at 4°C. The cleared lysate was incubated with pre-blocked anti-FLAG-M2 affinity gel (Sigma-A2220) overnight on a neutator at 4°C to allow binding of FH-Ago2. Next day beads were washed with IP buffer (20 mM Tris, pH 7.5, 5 mM MgCl$_2$, 150 mM KCl, 1× PMSF, 2 mM DTT) and were incubated with 300 nM purified recombinant HuR (wild type) or its mutant (Δ III, RRM III domain of HuR is deleted in this mutant) in an ASSAY buffer containing 20 mM Tris, pH 7.5, 5 mM MgCl$_2$, 150 mM KCl, 1× PMSF, 2 mM DTT, and 40 U/ml RNase inhibitor at 30°C for 60 min with shaking. For background counting, one set of reaction was set at 4°C for 0 min. After the reaction, Ago2 containing beads were separated by centrifugation at 2,000 × g for 2 min and the cleared supernatant was further centrifuged at 2,000 × g for 2 min and was added to HuR antibody bound (1:50) Protein G Agarose beads (pre-blocked with 5% BSA) in lysis buffer (20 mM Tris–HCl pH 7.5, 5 mM MgCl$_2$, 150 mM KCl, 1% Triton X-100,

1× PMSF, 2 mM DTT, 40 U/ml RNase inhibitor) and incubated overnight on a neutator at 4°C. Beads were washed with IP buffer and divided into two parts. RNA was isolated from one part and associated miRNA levels were measured by qRT–PCR. Relative level of proteins was analyzed by Western blot analysis using respective antibodies.

### TUNEL assay

TUNEL assay was performed using Promega DeadEnd™ Fluorometric TUNEL Assay kit for Fed, Starved, HA-HuR and pCIneo-expressing and DNase-treated Huh7 cells, as per manufacturer's protocol.

### Animal experiments

For the starvation experiment in adult BALB/c mice, Fed BALB/c mice were subjected to normal chow diet (containing amino acid, carbohydrate, and salt) and water, whereas sugar cubes and saline water were given to Starved BALB/c mice for 16 h. Animals were then sacrificed. For the expression of HA-HuR in BALB/c mice, adult BALB/c mice were injected with 25 μg pCIneo-HA-HuR-expressing plasmids and pCIneo (control)-expressing plasmids, respectively, via tail vein. Endotoxin-free plasmids were isolated from bacterial cell pellet using Sure Prep Plasmid Endofree Maxi kit (cat no. #NP-15363). Animals were sacrificed on the fourth day after injection. Liver and lung lysates were prepared in 1× RIPA buffer (25 mM Tris–HCl pH 7.6, 150 mM NaCl, 1% NP-40, 1% sodium deoxycholate, 0.1% SDS) by sonication followed by centrifugation at $20,000 \times g$, at 4°C for 1 h, and relative level of endogenous HuR was detected by Western blot analysis using anti-HuR antibody (starvation experiment). Similarly, exogenous HuR expression was checked by Western blot analysis using anti-HA antibody. β-Actin was used as the loading control. For isolation of RNA from tissues, Trizol reagent was used. For analysis of EV-associated RNA, serum fraction of blood was used. Relative levels of miRNA and mRNA in serum and tissues were quantified by qRT–PCR.

### Double-immunoprecipitation experiment

Huh7 cells were co-transfected with Flag-HuR (1 μg/6 well) and pRK5-HA-Ub (1 μg/6 well) plasmids in a six-well format using Lipofectamine 2000 following manufacturer's protocol. Cells were split after 24 h of transfection and were re-seeded in four 90-mm plates. After 24 h, cells of all four plates were subjected to starvation. Cells were harvested and lysed after 5 h of starvation, in the lysis buffer (20 mM Tris, pH 7.5, 5 mM MgCl₂, 150 mM KCl, 1% Triton X-100, 1% sodium deoxycholate, 1× PMSF, 2 mM DTT, 40 U/ml RNase inhibitor) for 30 min on a neutator at 4°C, followed by centrifugation at $3,000 \times g$ for 10 min at 4°C. The cleared lysate was incubated with pre-blocked anti-FLAG-M2 affinity gel (Sigma-A2220) overnight on a neutator at 4°C to allow binding of FLAG-HuR. Next day, beads were washed with IP buffer (20 mM Tris, pH 7.5, 5 mM MgCl₂, 150 mM KCl, 1× PMSF, 2 mM DTT) and were eluted using Flag-tripeptide-containing elution buffer (150 ng/μl Flag-tripeptide-eluent solution, 20 mM Tris, pH 7.5, 5 mM MgCl₂, 150 mM KCl, 1× PMSF, 2 mM DTT, 40 U/ml RNase inhibitor) by incubating on ice for 30 min with agitation after every 5 min. Then, the beads

were separated by centrifugation at $6,000 \times g$ for 30 s at 4°C. The supernatant or eluting solution was carefully collected and diluted with IP buffer (containing 40 U/ml RNase inhibitor) to increase its volume 2.5 times. One-fourth of the eluting solution was added to HuR antibody (1:50) bound Protein G Agarose beads, and 3/4th of the eluent solution was added to HA antibody (1:50) bound Protein G Agarose beads, respectively, both (pre-blocked with 5% BSA) in lysis buffer (20 mM Tris–HCl pH 7.5, 5 mM MgCl₂, 150 mM KCl, 1% Triton X-100, 1× PMSF, 2 mM DTT, 40 U/ml RNase inhibitor) and incubated overnight on a neutator at 4°C. Beads were washed with IP buffer and divided into two parts. RNA was isolated from one part, and associated miRNA level was measured by qRT–PCR. Relative level of HuR was analyzed by Western blot analysis using anti-HuR antibody.

### Cell senescence assay

Cells were assayed for senescence using Senescence Cells Histochemical Staining Kit (CS0030-1KT) from Sigma. Briefly, cells were assayed for senescence after fixation of the cells in a buffer containing 20% formaldehyde, 2% glutaraldehyde, 70.4 mM Na₂HPO₄, 14.7 mM KH₂PO₄, 1.37 M NaCl, and 26.8 mM KCl as a stock 10% solution, for 7 min. Following fixation, the cells were stained with a buffered staining solution containing 5 mM potassium ferricyanide and 5 mM potassium ferrocyanide along with 1 mg/ml of X-Gal solution at pH 6.0. The cells were kept at 37°C for overnight, following which the cells were thoroughly washed with PBS and finally mounted on a slide with Vectashield DAPI (H-1200, Vector Laboratories) for observation with a Nikon Eclipse Ti microscope equipped with 10× Plan Fluor 10×/0.30 objective. Images were captured using a Nikon Ri1 camera.

### Statistical analysis

All graphs and statistical analyses were generated in GraphPad Prism 5.00 (GraphPad, San Diego, CA, USA). Nonparametric unpaired $t$-test was used for analysis. $P$-values $< 0.05$ were considered to be statistically significant and $> 0.05$ were not significant (ns). Error bars indicate mean $\pm$ s.e.m.

### Ethics statement

All procedures were performed in accordance with a protocol approved by the Institutional Animal Ethics Committee (IAEC; Approval No 147/1999/CPCSEA Ref SNB/2011). All the experimentations were performed according to the National Regulatory Guidelines issued by the Committee for the Purpose of Supervision of Experiments on Animals, Ministry of Environment and Forest, Govt. of India. All the experiments involving animals were carried out with prior approval of the institutional animal ethics committee.

### Post-imaging analysis and others

All Western blot and Northern blot images were processed with Adobe Photoshop CS4 for all linear adjustments and cropping. All images captured on Nikon Eclipse Ti microscope were analyzed and processed with Nikon NIS ELEMENT AR 3.1 software. Image cropping was done using Adobe Photoshop CS4.

### Preparation of recombinant HuR and HuRΔIII

For expression of HuR and mutant proteins in *E. coli*, the gene was cloned in pET42a(+) (Novagen) between NdeI and XhoI sites. The protein was expressed in *E. coli* BL21DE3 codon Plus expression competent bacterial cells. Overnight cultures of *E. coli* BL21 transformed with plasmids expressing HuR or its mutants were diluted at 1:200 with the LB medium. At A600 of 0.3, cultures were induced with IPTG (0.5 mM) and grown overnight at 18°C. Cells were spun down and lysed by incubation with lysis buffer [20 mM Tris–HCl, pH 7.5, 300 mM KCl, 2 mM $MgCl_2$, 5 mM β-mercaptoethanol, 50 mM imidazole, 0.5% Triton X-100, 5% glycerol, 0.5 mg/ml lysozyme, 1× EDTA-free protease inhibitor cocktail (Roche)] for 30 min followed by sonication (10 s, three pulses, 80% power).

The lysate was centrifuged at $16,000 \times g$ for 15 min at 4°C. The supernatant was incubated with pre-equilibrated Ni-NTA Agarose beads (Qiagen) for 4 h at 4°C. The beads were washed twice by incubating with wash buffer (20 mM Tris–HCl, pH 7.5,150 mM KCl, 2 mM $MgCl_2$, 5 mM β-mercaptoethanol, 50 mM imidazole, 0.5% Triton X-100, 5% glycerol) with rotation, on neutator, each for 10 min, at 4°C.

His-tagged proteins were eluted by incubating beads with different elution buffers (20 mM Tris–HCl, pH 7.5,150 mM KCl, 2 mM $MgCl_2$, 5 mM β-mercaptoethanol, 0.5% Triton X-100, 5% glycerol) of different and increasing imidazole concentrations, each for 15 min at 4°C with rotation on the neutator. The purity of the eluted protein was checked by Coomassie staining, and the eluted protein was dialyzed against storage buffer (20 mM Tris–HCl, pH 7.5, 150 mM KCl, 2 mM $MgCl_2$, 0.5 mM DTT, 0.1% Triton X-100 and 8% glycerol) and stored at −80°C.

### RNA end-labeling

About 10 pmoles of synthetic RNA TNFα 3′UTR ARE RNA (34 nt) or miR-122 (22 nt) or miR-122* was incubated with 10× $T_4$ PNK buffer, DEPC-treated $H_2O$, $T_4$ PNK (10 units/μl) and γ-$^{32}$P-ATP (10 μCu/μl) at 37°C for 30 min without shaking. Reaction was stopped with Tris–EDTA solution and filtered through RNA column. RNA was extracted with Trizol LS and $CHCl_3$ and precipitated with isopropanol at −20°C for 2 h. RNA pellet was resuspended in DEPC-treated $H_2O$ at a final concentration of 500 fmoles/μl.

### EMSA assay

RNA Mix was prepared with 100 fmoles of γ-$^{32}$P-ATP-end labeled miR-122, miR-122* or TNFα 3′UTR ARE RNA in EMSA buffer (20 mM Tris–HCl, pH 7.8, 2 mM $MgCl_2$, 150 mM KCl, 0.1% Triton X-100, 0.5 mM DTT and ultrapure water), heparin (5 mg/ml final conc.), BSA (1 mg/ml final conc.), and glycerol (10% final conc.). Recombinant HuR proteins (FL and ΔIII) were diluted in EMSA buffer to desired conc. and incubated with the RNA mix on ice for 15 min. Equal volume of 2× EMSA dye (20% glycerol, 20 mM Tris–HCl, pH 7.8, 0.05% bromophenol blue and 0.05% xylene cyanol and water) was added to reaction mix and run on 6% Native PAGE in 0.5× TBE at 4°C at 6 mA constant current. Gel was dried and PhosphorImaging of the blots was performed in CyclonePlus Storage Phosphor System (PerkinElmer).

### *In vitro* transcription

HCV-JFH1 RNA was transcribed *in vitro* from the XbaI linearized HCV-JFH1 plasmid DNA construct using T7 RNA polymerase (Thermo Scientific). The transcription reaction was carried out under standard conditions using 2.5 μg of linear template DNA at 37°C for 4 h. After alcohol precipitation, the RNA was resuspended in nuclease-free water and used for transfection.

### *In vitro* ubiquitination assay

HA–ubiquitin expression plasmid-transfected Huh7 cells grown in two 90-mm plates were lysed in hypotonic buffer (10 mM Tris,pH 7.5, 1 mM $MgCl_2$, 1 mM DTT) by sonication and the lysate was centrifuged at $20,000 \times g$ at 4°C for 15 min. Supernatant was collected and adjusted to make the cell lysate in 1× ubiquitination buffer (1× UB) containing 50 mM Tris pH 7.8, 5 mM $MgCl_2$, 0.1% Tween-20, 1 mM DTT, 2 mM ATP.

Cell lysate in 1× UB was incubated with purified recombinant his-tagged-HuR protein (wild type) in presence of 1× UB, 0.5 mM ATP, 20 μM MG132 at 25°C for 1 h. This reaction mix was diluted with 1× lysis buffer (20 mM Tris, pH 7.5, 5 mM $MgCl_2$, 150 mM KCl, 1% Triton X-100, 1× PMSF, 2 mM DTT) and incubated with anti-6-HIS antibody bound (1:50) pre-blocked Protein G Agarose beads overnight with rotation at 4°C. Beads were washed with 1× immunoprecipitation buffer (20 mM Tris, pH 7.5, 5 mM $MgCl_2$, 150 mM KCl, 1× PMSF, 2 mM DTT) and Western blotted for HA, 6-HIS, HuR-specific antibodies respectively.

### mirVANA miRNA and rHuR-binding assay immunoprecipitation experiment

miRNA pool of Huh7 M cells was isolated by *mir*VANA miRNA Isolation Kit (Life Technologies) as per the manufacturer's protocol. This miRNA pool was incubated with purified recombinant his-tagged-HuR protein (wild type) in presence of 1× EMSA buffer, 10% glycerol, 5 μg/μl heparin, 1 μg/μl BSA on ice for 1 h. After reaction, 1× IP buffer was added and this diluted reaction mixture was incubated for 2 h with anti-HuR-bound (1:50) pre-blocked Protein G agarose beads with rotation at 4°C. Beads were washed with 1× IP buffer and Western blotted for HuR-specific antibody. Levels of various HuR-bound miRNAs were quantified by real-time qRT–PCR and compared against respective miRNA levels in input samples.

### Additional methods and reagents

Sources of plasmids and DNA templates for PCR amplification, sources of antibodies, sources of siRNAs, miRNAs, antisense oligonucleotides are described in the Tables EV1, EV2 and EV3.

**Expanded View** for this article is available online.

### Acknowledgements

We would like to thank Witold Filipowicz, Siddhartha Roy, Dhrubajyoti Chattopadhyay and Samit Chattopadhyay. We also convey our thanks to P Chakraborty, N Ali, M Sen, R Pillai, and P Nath for their generous help with reagents, plasmids constructs, and animal experiments. We are particularly indebted to Witold Filipowicz for his critical comments and suggestions. The

work has been primarily funded by The Wellcome Trust ISRF Fund (084324/Z/07/A) received by SNB and was also supported by Lady Tata memorial Trust Young Researcher Award Fund and the HFSPO Career Development Award Fund (CSA-25/2007-C) to SNB. All the authors were supported by fellowship either from CSIR or from The Wellcome Trust. SD acknowledges the support of JC Bose Fellowship.

## Author contributions

KM and SNB conceived and designed the experiments and wrote the manuscript. KM, BG, and SG performed all the experiments. YC helped in animal experiments. SD and SNB designed and SS performed the HCV-related experiments described in Fig EV3. Experiments in Fig EV5A and B are done by SNB.

## Conflict of interest

The authors declare that they have no conflict of interest.

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
