## [Review Process File · EMBO Reports]

Manuscript EMBO-2015-41930

Reversible HuR-microRNA Binding Controls Extracellular Export of miR-122 & Augments Stress Response

Kamalika Mukherjee, Bartika Ghoshal, Souvik Ghosh, Yogaditya Chakrabarty and Suvendra Bhattacharyya

Corresponding author: Suvendra Bhattacharyya, CSIR-Indian Institute of Chemical Biology

Review timeline:

Submission date:	17 December 2015
Editorial Decision:	12 January 2016
Revision received:	22 April 2016
Editorial Decision:	13 May 2016
Revision received:	08 June 2016
Accepted:	09 June 2016

Editor: Esther Schnapp

Transaction Report:

1st Editorial Decision

12 January 2016

Thank you for the transfer of your manuscript to EMBO reports. We have now received the comments from the referees, which are copied below.

As you will see, while the referees acknowledge that the findings are potentially interesting and novel, they also all point out that the data are insufficient to support the main conclusions. The main concerns are whether the miRNAs might be degraded instead of exported in response to amino acid starvation, that the technical quality of the study is low with a number of missing controls, quantifications and statistics, that the evidence for HuR ubiquitination is too weak, and that the biological relevance of the findings remains unknown.

From these comments it is clear that publication of the manuscript in our journal cannot be considered at this stage. On the other hand, given the potential interest of the findings, I would like to give you the opportunity to address the concerns and would be willing to consider a revised manuscript with the understanding that the referee concerns must be fully addressed and their suggestions taken on board. Upon cross-commenting on each others' reports, referee 3 indicates that points 12 and 13 by referee 2 are out of the scope of this study and would not need to be addressed, however, better evidence for HuR ubiquitination must be provided. It would also not need to be addressed how mRNA targets affect miRNA secretion (referee 1 point 3).

Please address all referee concerns in a complete point-by-point response. Acceptance of the manuscript will depend on a positive outcome of a second round of review.

REFEREE REPORTS

Referee #1:

The mechanisms that underpin selective turnover of miRNAs is still poorly understood. In the paper by Mukherjee et al cells are stressed by amino acid starvation and examined for the possible effects on miRNA sorting into exosomes and the possible role of the HuR protein. The authors claim that HuR replaces miRNPs from target messages and is both necessary and sufficient for the export of corresponding miRNAs. During this process, HuR may reversibly bind miRNAs replacing them from AGO2 on the ER. HuR seems to get liberated from associating miRNAs upon ubiquitination at or near MVBs, possibly explaining their sorting into exosomes. Subsequently HuR-unloaded miRNAs are exported from the cell via MBV-derived EVs (presumably exosomes) thereby reducing cellular miR-122 levels during amino acid starvation.

The authors conclude that by modulating extracellular export of miR-122, HuR controls stress response in amino acid starved human hepatic cells. The authors propose a model how HuR may impact miRNA turnover during starvation. The rules of turnover with respect to sorting and secretion of miRNAs (that are functional in exosomes) is an important field of study and the current paper by Mukherjee, provide potential interesting novel insight. However, important pieces of information are missing, especially compared to what has been published. Additional experimentation needs to be done to better assess the biological relevance of the current findings and substantiate the claims.

Major Points;

This is a key point as it is unclear whether export of miR-122 is the major contributor to the reduction of intracellular miR-122.

1) While I can imagine that miR-122 transcription is not affected during starvation (although this was not formally shown) what about 'degradation'? Experiments with nascent RNA could provide additional clues as to the biology of miR-122 in cells and exosomes. It was also reported that coincident with degradation, and while still bound to Argonaute, targeted miRNAs are 3' terminally tailed and trimmed, this should be easily picked up within a small RNA-seq experiment.

2) While the data shown is informative, sequencing of exosomes and cells under starvation condition should provide additional, if not crucial, clues into miR-122 turnover during starvation, as PCR will mainly detect mature (canonical) miR-122. Recent studies suggested that 3'-end modifications (non-templated additions) of small-RNAs may drive expulsion (Koppers-lalic et al., Cell Reports 2014). Can the authors confirm or exclude a role for such modifications in their model?

3) In another paper (Squadrito et al., Cell Reports 2014) it has been suggested that target mRNA levels impact miRNA secretion. Would varying ALDO-A or CAT mRNA levels, known and biologically relevant targets, have an effect on sorting and secretion? Luciferase (miRNA-122 activity) reporter-assays with 3'UTR and mutated binding sites of targets would yield better insight into the functional relevance of HuR sequestering of miR-122. A relevant control 3'UTR of a miRNA that is not prone to secretion via exosomes should serve as control.

4) The biological relevance of the story is hard to interpret. Luna et al (Cell 2015) showed an interesting link between miR-122 and sequestering by HCV RNA interfering with AGO binding supporting replication. This work should at least be discussed or ideally, HCV-RNA target should be taken along with experiments providing more biological and functional relevance of this in potential interesting paper.

Minor points;

-The quality of blots is sometimes poor

-Fig1H, there is no CD63 in the HEP cells?

-The text with the figures is not easily readable in many figures.

-The authors claim that the cellular level of miRNAs increase upon starvation and represent this in

relative numbers. Raw (normalized) ct-values would provide a better insight.

-It would behoove the authors to show miR-122 copy-numbers/cell this could give an indication on the functional relevance next-to or as a weaker alternative for reporter (activity) assays.

Referee #2:

The authors in this manuscript suggest that amino acid starvation of liver cells leads to the increased secretion of miR-122 in Extracellular Vesicles (EVs). They furthermore suggest that the loading of this miRNA in these vesicles occurs due to the HuR-mediated displacement of Ago2 from the small noncoding RNA. They, lastly, suggest that the ubiquitination of HuR frees it from miR-122 in these vesicles.

The notion that HuR could be involved in a process that mediates the loading of miRNAs in EVs is very interesting and intriguing. The authors have however not provided sufficient experimental evidence that convincingly supports the premise of their idea. The data that is provided to support this model is generally weak or of poor quality. Indeed, there are numerous instances within the manuscript where the quality of the provided data or the lack of adequate controls renders it impossible to judge if the conclusions drawn by the authors are indeed valid. Another major concern is that the authors have failed to mechanistically describe how HuR is involved in the loading of this miRNA in these EVs and if indeed the ubiquitination of HuR is indeed important in this process. These, as well as other concerns, are described below:

- 1) The western blot provided in Fig. 1C is of very poor quality. It is difficult to visualize, from this western, that HuR levels in EVs are actually increased in starved cells. The author should, therefore, provide a better quality western that more accurately reflects the result presented in the plot. Additionally, the authors should include a negative control (protein absent from EVs) in the Western in order to efficiently demonstrate the purity of the isolated vesicles.
- 2) The results presented in Fig. 1G are not convincing. The enhancement of miR-122 levels is only seen in fraction 9 although both fractions 9 and 10 both appear to have CD63 positive EVs. Can the authors explain why this is the case? Additionally, the provided data should contain statistical analyses in order to strengthen the conclusion that the levels of miR-122 are indeed significantly increased in these EV positive fractions.
- 3) Statistical analyses should be provided in Fig. 1I in order to demonstrate the significance of the results. This concern also applies to all other bar graphs in the manuscript lacking such analyses.
- 4) The authors should include, in Fig. 2A, an IgG negative control antibody in these experiments. Furthermore the results shown in this Figure do not convincingly demonstrate that HuR directly interacts with miR-122. Indeed, it is possible that this interaction occurs in a mRNA dependent manner due to their individual association with common mRNA targets such as the cat-1 mRNA. Additionally evidence should be provided to show that this interaction does indeed occur in a direct manner.
- 5) In Fig. 2D the authors should demonstrate how the knockdown of HuR affects the levels of miR-122 associated with EVs isolated from starved Huh7 cells.
- 6) In Fig. 2E the authors should demonstrate if the effect of knocking down HuR on cat-1 mRNA levels is due to increased levels of cellular miR-122. Are these effects negated if the cells are transfected with anti-miR-122 antagomirs?
- 7) The western blot shown in Figure 2E does not convincingly demonstrate, as suggested by the authors, that the knockdown of HuR decreased the phosphorylation of eIF-2 α .
- 8) The western blot in Fig. 3A is lacking a loading control to allow one to assess the extent to which HuR is overexpressed in these cells.
- 9) The authors conclude, from their studies, that HuR displaces miR-122 from Ago2 leading to the increased association of this small noncoding RNA in the EVs. In order to strengthen this conclusion the authors should investigate, in Fig. 3B, if the overexpression of Ago2 can negate the increased association of miR-122 with EVs due to the overexpression of HA-HuR. Does the knockdown of Ago2 increase the association of miR-122 with EVs isolated from both fed and starved cells?
- 10) The western blots presented in Fig. 4C and E are of poor quality and must be replaced with better quality images which better reflects the conclusions drawn by the authors from these experiments.

11) The data provided in Fig. 5C, as stated above, does not negate the possibility that the association of HuR with miR-122 occurs indirectly due to their individual association with independent cis-elements in the cat-1 mRNA 3'UTR. Additional evidence should be provided to show that the HuR mediated displacement of Ago2 associated with miR-122 does indeed occur due to a direct interaction with this small noncoding RNA.

12) Although the data provided do indeed suggest that HuR affects the association of miR-122 with EVs the authors have failed to explain, mechanistically, how HuR mediates the loading of miR-122 into these vesicles. What is the role of HuR in the loading of these small noncoding RNAs into these EVs?

13) In Figure 6 the authors should identify the HuR residue which is ubiquitinated in these cells and, furthermore, assess how a HuR mutant for this ubiquitinated site affects miR-122 levels in these vesicles.

Referee #3:

The paper documents the impact of HuR on levels of miR-122 in cells and exosomes both in cell culture and in mice. Similar effects are observed in starved cells and mice however the dependence of starvation effects on HuR are not strictly defined. While data on the effects of starvation and HuR on miR-122 content of exosomes and cells is acceptable with a few additional controls there are more significant issues with data on ubiquitination of HuR and its binding to miRNA that is used to develop a model of how HuR affects levels of miR-122 in exosomes and cells.

1. Fig.1A- is there a size change in miR-122 in starved cells? There appears to be a slight downward shift.
2. Provide evidence for specificity of HuR antibody. Blots are often messy.
3. Test for levels of CAT1 and aldolase in exosome fractions of starved and fed to see if mRNA comes with miR-122 into exosomes. Mechanistically important.
4. SMPD2 siRNA decreases release of miR-122 release in exosomes by only about 50%. Not all exosome populations are likely affected. In starved cells miR-122 release increases and SMPD2 siRNA brings this back to the same level as in fed cells. This suggests that while not all exosomes released are SMPD2 siRNA dependent (or 50% of miR-122 is not in exosomes), the exosomes released in response to starvation are SMPD2-dependent. The Nanosight data in Supp. Fig.1 also suggests multiple size peaks in EV preparations. Which subpopulations by size are SMPD2-dependent, and affected by starvation and contain miR-122. Could be lipid particles or other types of vesicles that contain miR-122. Similarly, comparing Fig.3B and E there are large increases in miR-122 in EV in Fig.3B, but a small increase in CD63+ exosomes isolated with beads in 3F, and a small decrease in miR-122 in EV preparations when exosome release is blocked with GW4869. This again suggests that the majority of miR-122 release in EV preps is not in exosomes.
5. Show that SMPD2 siRNA also decreases miR-122 in cells and increases CAT-1.
6. Starvation induces metabolic signaling pathways and autophagy. Autophagy can degrade RNAs. The authors need to show that disappearance of miRNA in starved cells is not due to autophagy using genetic or siRNA tools, not drugs. Since the amount of exosomes released from cells is so small (ug vs. ug) compared to cells, autophagic degradation rather than exosomal release would seem a much more plausible mechanism for decreased miR-122 levels in cells. At minimum the authors should provide RNA copy number data for exosomes vs. total cell RNA to demonstrate that miR-122 amounts found in exosomes could account for those disappearing from cells.
7. In 1G gradients data that miR-122 levels increase in exosome fractions of starved cells is very weak - the effect is weak in an experiment with lots of apparent variability. Better evidence is required. Indeed, this suggests there are important amounts of miR-122 in non-exosomal parts of the EV preparations.
8. To show that starvation requires HuR for increased export of miR-122 in EVs the authors need to include data in fed +control siRNA/HuR siRNA, starved +control siRNA/HuR siRNA for miR-122 levels and CAT-1 levels in EVs and cells.
9. Fig.1H. Is there a statistically significant increase in TUNEL positive cells in starved cells. Large positive control bar makes it difficult to see differences in test samples. Stats required.
10. Fig.2A. Blot for HuR is over-exposed. Show lesser exposure of HuR blot and quantify changes in HuR levels in cells and IPs used for RT-qPCR. Changes in HuR levels could account for changes in HuR bound miRNAs in IPs.

11. Purported differences in Fig.2F are too small to be credibly published without further analyses such as quantifications of eIF2a-P and eIF4ebp1 over multiple experiments or assays of larger effects on downstream effectors. A scrambled anti-miRNA control should be used rather than anti-let-7. At present this data should be eliminated from the paper.
12. All bar graphs require statistical analysis. These are missing in at least Fig.2C, 3E, 5E.
13. In all experiments using IPs of proteins followed by measuring RNA levels, blots for the immunoprecipitated proteins are required to show that similar amounts of proteins were immunoprecipitated in each sample. E.g. Fig.5A, 2A.
14. Fig.5C. miRNA associates with HuR after incubation with AGO2-miRNA complexes. Is this specific to HuR or would the same result occur with any protein. Try experiment with a control RNA binding protein.
15. Why does levels of CD63 in EV preps consistently decrease in cells transfected with HuR? Fig.3H, 3D
16. In many places the lack of molecular mass markers and the messy blots with multiple bands makes it very difficult to understand what is happening. This is particularly bad in Fig.6 on ubiquitination, but also in Fig.5E IP HuR blot for AGO2. The lower band for AGO2 in the IP on the right side of the figure aligns with a lower band for AGO2. Is this a variant of AGO2? Show molecular weight markers.
17. Fig.5C. miRNA associates with HuR after incubation with AGO2-miRNA complexes. Is this specific to HuR or would the same result occur with any protein. HuR needs to be compared to a control RNA binding protein that is not known to bind miRNA.
18. Data showing ubiquitination of HuR at MVB is not acceptable. There is no definitive evidence for HuR ubiquitination, let alone its ubiquitination at MVB. The major issues are:
 - (a) Necessary controls are lacking in most or all of the IPs (control antibody IP, IP of HuR in cells not-transfected with Ub-HA). This renders it impossible to make conclusions from the data. In most or all of the blots it is unclear which bands are HuR alone or HuR-Ub, or some non-specific band or other splicing or modification of HuR. For example, Fig.5D On right, band for Flag-HuR is already detected with ubiquitin antibody. Does this mean all HuR is ubiquitinated? This seems unlikely. Lack of molecular mass markers in Fig.6 makes it very difficult to understand blots in Fig.6 and elsewhere.
 - (b) One expects a ladder and/or smear for polyubiquitination of a protein, not distinct bands. Fig.6C need to show blots for HuR, not just HA (Ubiquitin) and should show top of gel.
 - (c) It is not clear that HuR is ubiquitinated at MVB in normal conditions without over-expression of HA-Ub. The authors should use an antibody to endogenous ubiquitin (without transfecting Ub-HA)
19. Role of hinge region of HuR in export of miR-122 in EVs. This could be interesting but controls are missing including validation that HuR is not ubiquitinated in their systems when lacking the hinge region and validation that HuR lacking the hinge retains miR-122 binding capacity.

Minor Comments

1. For starvation cells are incubated with Hank's with dialyzed FCS. Is this also exosome-depleted by centrifugation. What is the dialysis cut-off? The authors refer to their starvation method as amino-acid starvation. This is inaccurate as cells are starved of many things (Hanks vs. DMEM and dialyzed FCS).
2. Something strange is happening with miR-21 in cells. Like miR-122 its binding to HuR increases in starved cells and its levels in EVs are affected. However, miR-21 levels in cells are unaffected. Several explanations are possible. The authors should mention this exception.
3. Missing blot confirming increased HuR-HA expression in Fig.3H.

1st Revision - authors' response

22 April 2016

Please find the revised version of the manuscript EMBOR-2015-41930V2 now titled as "Ubiquitination Restricts miRNA Binding of HuR and Augments Extracellular Export of miR-122 in Stressed Hepatic Cells" by K. Mukherjee et al., for publication in *EMBO Reports*. We consider the changed title is a more representative one for the key aspects that the revised manuscript conveys.

In this revised manuscript, we have addressed all the concerns the reviewers had against its previous version. We have incorporated several new experiments in support of our claim that human ELAVL1 protein HuR acts as a facilitator of extracellular export of miRNAs. We found HuR, an

ubiquitously expressed RNA binding protein, binds miRNAs in a reversible manner on endoplasmic reticulum while ubiquitination and miRNA unloading of HuR happens on Multivesicular bodies (MVBs) that ensure subsequent Extracellular Vesicles (EVs) mediated release of unloaded miRNAs in extracellular space.

We have addressed all the major and minor criticisms that the reviewers had related to the quality of the data presented in the previous version of this manuscript. We also have added proper controls in the experiments as they were initially missing for some of the data presented earlier. The experiments added in the revised manuscript have strengthened the importance of ubiquitination of HuR for exosome driven export of miRNAs in mammalian cells. We also added new experiments to show how HuR driven EV-mediated export of miRNA controls stress response and cell growth in mammalian hepatic and breast cancer cells respectively. That, we hope, will satisfy the reviewers who do have concerns related to the physiological significance of HuR driven export of miRNAs. We also have experiments in the revised version to suggest EV mediated export rather than cellular degradation as the major mechanism of lowering miR-122 in amino acid starved hepatic cells. We have our detailed response against each point raised by the reviewers in the accompanying "Response to Reviewers Comments"

Response to Reviewer's Comments:

Reviewer 1

Major Points:

This is a key point as it is unclear whether export of miR-122 is the major contributor to the reduction of intracellular miR-122.

1) While I can imagine that miR-122 transcription is not affected during starvation (although this was not formally shown) what about 'degradation'? Experiments with nascent RNA could provide additional clues as to the biology of miR-122 in cells and exosomes. It was also reported that coincident with degradation, and while still bound to Argonaute, targeted miRNAs are 3' terminally tailed and trimmed, this should be easily picked up within a small RNA-seq experiment.

We agree with the reviewer that targeted modification of miRNAs could make it degradable or exportable therefore assessing the modification of miRNA under Fed and Starved condition would be a good suggestion. We have performed the Northern blot after running the RNA from Fed and Starved cells in 15% Urea PAGE for analyzing the miR-122 and to distinguish it from species of having single base addition or deletion, which is otherwise impossible with RT-qPCR. However, we did not observe a change in the length of the miR-122 either in cellular or in exosomal pool upon starvation (Fig. 1A). This finding rules out the possibility of modification of miR-122 as the primary signal for its degradation or export in starved cells.

With starvation we also did not observe an accompanying decrease in the level of the precursor pre-miR-122 (Fig. 1C). This neglects contribution of reduced transcription in downregulation of miR-122 in starved hepatic cells. It may be noted that when GW4869 was applied to specifically block exosomal export of miRNA, we did not observe a significant change in cellular miR-122 level upon starvation (Fig. 1E). This suggests limited or no-contribution of non-exosomal degradation pathways for starvation induced lowering of miR-122 in hepatic cells. We have also tried experiments with alpha-amanitin to block *de novo* production of RNAs in GW4869 treated cells and documented similar reduction in mature miRNA levels in both Fed and Starved condition (Fig. EV1E). This further strengthens the claim that exosomal export is the major contributor in lowering of miR-122 in Starved Huh7 cells.

2) While the data shown is informative, sequencing of exosomes and cells under starvation condition should provide additional, if not crucial, clues into miR-122 turnover during starvation, as PCR will mainly detect mature (canonical) miR-122. Recent studies suggested that 3'-end modifications (non-templated additions) of small-RNAs may drive expulsion (Koppers-lalic et al., Cell Reports 2014). Can the authors confirm or exclude a role for such modifications in their model?

This is also a good suggestion and we thank this reviewer for citing this important reference in this connection that we missed earlier. But as our manuscript focuses on miR-122 export in Starved cells and we did not observe a change in its length (evident from Northern Blot data in Fig 1A), we

preferred not to explore the changes in modification status of other miRNAs in Starved and Fed cells as we find it will defocus the present manuscript. Rather we cited this important reference in the discussion part of revised manuscript and cited the possibility that modified miRNAs could be a better substrate for HuR/non-HuR mediated export where they are specifically excluded or preferentially taken on board during the export process.

3) In another paper (Squadrito et al., Cell Reports 2014) it has been suggested that target mRNA levels impact miRNA secretion. Would varying ALDO-A or CAT mRNA levels, known and biologically relevant targets, have an effect on sorting and secretion? Luciferase (miRNA-122 activity) reporter-assays with 3'UTR and mutated binding sites of targets would yield better insight into the functional relevance of HuR sequestering of miR-122. A relevant control 3'UTR of a miRNA that is not prone to secretion via exosomes should serve as control.

We agree with Referee 3 that exploration of this point will be out-of context and therefore we opted for not doing in-depth exploration of this point. Rather we added a paragraph in the discussion where we pointed out how target availability could act as a possible intrinsic factor that may modify the export of miR-122 in starved mammalian cells. In this context we cited the Reference mentioned above and discussed it.

4) The biological relevance of the story is hard to interpret. Luna et al (Cell 2015) showed an interesting link between miR-122 and sequestering by HCV RNA interfering with AGO binding supporting replication. This work should at least be discussed or ideally, HCV-RNA target should be taken along with experiments providing more biological and functional relevance of this in potential interesting paper.

We have tried the experiment in Huh7 cells expressing the HCV replicon and co-expressing Myc-HuR to score the changes in miR-122 level. As expected we documented a reduction in the level of miR-122 in presence of Myc-HuR but it was not accompanied by an increase in HCV replicon RNA. This is not surprising as HuR is also known to have a positive effect on HCV replicon level in Huh7 cells (Shwetha et al., J Virol. 2015). As HuR has opposite effects on HCV RNA and miR-122, effect of miR-122 depletion on HCV will be masked by positive effect of HuR on HCV replicon. We added this data in Fig EV3 and stated the findings in the discussion part.

As additional evidence strengthens a physiological relevance of exosomal export of miR-122, we have found that cells depleted for HuR showed a reduced stress response and reduced autophagy under starvation that get reversed with anti-miR-122 treatment (Figs 2E-F and EV2). In human breast cancer cell MDA-MB-231, let-7a expression increases senescence. We found HuR is responsible for exosomal export of let-7a in MDA-MB-231 cells that ensure cell proliferation. We have also documented an increase in senescence in MDA-MB-231 cells depleted for HuR (Fig. EV3).

Minor points;

-The quality of blots is sometimes poor

We have tried to improve the blot quality by trying the fresh western blot experiments wherever applicable

-Fig1H, there is no CD63 in the HEP cells?

This was not the case but the detection was difficult. We repeated the experiment with increased amount of protein to detect CD63 in Huh7 cells (Fig. 1K).

-The text with the figures is not easily readable in many figures.

We have paid attention to those panels and proportionately increased the text size, wherever necessary.

-The authors claim that the cellular level of miRNAs increase upon starvation and represent this in relative numbers. Raw (normalized) ct-values would provide a better insight.

As we had to use different amplification platforms such as ABI 7500, QuantStudio or BioRad Real time systems, the Ct values between different experimental groups should not be used for a comparison. Therefore relative quantification between control and experimental sets done individually for each experiment has been taken together for analysing the changes. It was adopted before and presented in the similar manner in both manuscripts we published on exosomal miRNA export (Ghosh et al., Mol. Biol. Cell, 2015 and Basu and Bhattacharyya, Nucleic Acids Res., 2014). Therefore, unless specifically required for, we prefer to go ahead with relative level representation as it will be easy to understand and follow the data by non-expert readers.

-It would behoove the authors to show miR-122 copy-numbers/cell this could give an indication on the functional relevance next-to or as a weaker alternative for reporter (activity) assays.

We have calculated the copy number of miR-122 and mentioned it in the discussion part. Please also see our response to Point 6 of reviewer 3.

Referee 2

1) The western blot provided in Fig. 1C is of very poor quality. It is difficult to visualize, from this western, that HuR levels in EVs are actually increased in starved cells. The author should, therefore, provide a better quality western that more accurately reflects the result presented in the plot. Additionally, the authors should include a negative control (protein absent from EVs) in the Western in order to efficiently demonstrate the purity of the isolated vesicles.

We agree with the referee that the western blot quality was inferior. But to get a good quality western blot of exosomal fractions for less abundant proteins is always a challenge. Nevertheless we repeated the experiment several times and present a western blot of improved quality to replace the old one. We have also done western blots for other positive and negative control proteins known to be present and absent in the EVs to demonstrate the purity of the isolates (Fig. 1D).

2) The results presented in Fig. 1G are not convincing. The enhancement of miR-122 levels is only seen in fraction 9 although both fractions 9 and 10 both appear to have CD63 positive EVs. Can the authors explain why this is the case? Additionally, the provided data should contain statistical analyses in order to strengthen the conclusion that the levels of miR-122 are indeed significantly increased in these EV positive fractions.

We have repeated the OptiPrep analysis and identified increased enrichment of miR-122 in CD63 positive fractions. The peak of CD63 present in the 9th fraction but whether exosome density varies with the cargos that they carry or whether there are heterogeneous population of exosomes existing in a supernatant is not clear from the existing literature. It is likely that exosomes also fuse among themselves to give bigger vesicles during isolation and storage. Therefore it will be very difficult to answer what will be characteristic of these CD63 positive but starvation driven miR-122 negative particles/vesicles present in the 10th fraction. We hypothesize that it could be exosomes of different class that are not primarily used to carry miRNAs. The cause and mechanism of possible exosome heterogeneity is certainly beyond the scope of this study. But, as we can't directly show that starvation expelled miRNAs is only carried by exosomes, we added this concern in the discussion part and prefer to use EVs rather than exosomes where it is appropriate.

3) Statistical analyses should be provided in Fig. 1I in order to demonstrate the significance of the results. This concern also applies to all other bar graphs in the manuscript lacking such analyses.

We have tried to provide the missing statistics for all the bar graph data we have presented in this manuscript.

4) The authors should include, in Fig. 2A, an IgG negative control antibody in these experiments. Furthermore the results shown in this Figure do not convincingly demonstrate that HuR directly interacts with miR-122. Indeed, it is possible that this interaction occurs in a mRNA dependent manner due to their individual association with common mRNA targets such as the cat-1 mRNA. Additionally evidence should be provided to show that this interaction does indeed occur in a direct manner.

We have included an IgG negative control as requested to confirm the specificity of anti-HuR antibody used for IP reaction (Fig. 2A). We did not get any miR-122 amplification from RNA isolated from control IgG IPed materials. We also have incorporated IP data in Fig. EV5B (*in vivo*) and Figs 5A and D (*in vitro*) where we did not obtain any interaction between HuR and Ago2 (similar result was also reported earlier by Kundu et al., *Nucleic Acids Res.* 2012). As miRNA binding to target RNA is mediated by Ago2-miRNA (miRNP) complexes, no detectable interaction between HuR and Ago2 rule out the possibility of miRNA HuR interaction through an indirect route. We further use recombinant and pure HuR and incubated it with single stranded miR-122 and document direct, concentration dependent binding of miR-122 to (Fig. 5E). The truncated mutant of HuR without the RRMIII showed reduced miR-122 binding consistent with its impaired miRNA displacement activity (Fig. 5D and E). This data strongly suggests a direct binding of HuR to miR-122.

5) In Fig. 2D the authors should demonstrate how the knockdown of HuR affects the levels of miR-122 associated with EVs isolated from starved Huh7 cells.

We have estimated the changes of miR-122 in the EVs isolated from starved Huh7 cells upon HuR knock down (Fig. 2D).

6) In Fig. 2E the authors should demonstrate if the effect of knocking down HuR on *cat-1* mRNA levels is due to increased levels of cellular miR-122. Are these effects negated if the cells are transfected with anti-miR-122 antagomirs?

The suggested experiment has already been done and presented in the Fig. 2F.

7) The western blot shown in Figure 2E does not convincingly demonstrate, as suggested by the authors, that the knockdown of HuR decreased the phosphorylation of eIF-2a.

We have repeated the experiment multiple times and have incorporated the quantified data and calculated the statistical significance from multiple data (Fig 2E) .A representative western blot is also shown in Fig. EV2B.

8) The western blot in Fig. 3A is lacking a loading control to allow one to assess the extent to which HuR is overexpressed in these cells.

We have introduced the loading control to give an idea of HuR overexpression in Fig. 3B.

9) The authors conclude, from their studies, that HuR displaces miR-122 from Ago2 leading to the increased association of this small noncoding RNA in the EVs. In order to strengthen this conclusion the authors should investigate, in Fig. 3B, if the overexpression of Ago2 can negate the increased association of miR-122 with EVs due to the overexpression of HA-HuR. Does the knockdown of Ago2 increase the association of miR-122 with EVs isolated from both fed and starved cells?

This was an excellent suggestion and we follow it to express FH-Ago2 and found that Ago2 over expression reduces exosomal miR-122 content when HA-HuR was also expressed. However, the cellular miR-122 level also drops upon HA-HuR and FH-Ago2 coexpression . The reverse experiment with siRNA mediated knock down of Ago2 in Huh7 cells was not ideal as different Agos may have redundant role to play. Additionally, in starved condition we already have an increased export of miR-122 and thus over expression rather than depletion of Agos is expected to have a strong effect. Therefore, it would have been interesting to test whether Ago expression may counter the Starvation dependent lowering of miR-122 in Huh7 cells. Unfortunately, we could not detect any significant change in cellular level of miR-122 in Starved cells upon Ago2 expression. This may be easily explained if HuR mediated uncoupling of miRNA from Ago2 is an irreversible process. This may also be due to an unidentified but additional level of regulation present in Starved cells (such as availability of miR-122 targets) that in conjugation of HuR may plays a role in expelling miR-122 which is independent of cellular Ago2 level. Direct interpretation of these results, we consider, are beyond the scope of the present study but considering its importance we have described the finding in the Discussion part (Fig EV5D).

10) The western blots presented in Fig. 4C and E are of poor quality and must be replaced with better quality images which better reflects the conclusions drawn by the authors from these experiments.

We have introduced the new WB data for Fig. 4C. But 4E, we could not replace it with a better blot as we faced the problem of protein degradation with frozen liver samples after thawing. Restriction with number of animal usage prevented us to redo the full experiment again for getting fresh sample for new WB. We request the reviewer to be sympathetic with us related to this issue.

11) The data provided in Fig. 5C, as stated above, does not negate the possibility that the association of HuR with miR-122 occurs indirectly due to their individual association with independent cis-elements in the cat-1 mRNA 3'UTR. Additional evidence should be provided to show that the HuR mediated displacement of Ago2 associated with miR-122 does indeed occur due to a direct interaction with this small noncoding RNA.

This is an important concern but we have already addressed this in our response against point 4 of the same reviewer.

12) Although the data provided do indeed suggest that HuR affects the association of miR-122 with EVs the authors have failed to explain, mechanistically, how HuR mediates the loading of miR-122 into these vesicles. What is the role of HuR in the loading of these small noncoding RNAs into these EVs?

Following the Editorial suggestions we preferred not to explore the point raised by the Reviewer 2.

13) In Figure 6 the authors should identify the HuR residue which is ubiquitinated in these cells and, furthermore, assess how a HuR mutant for this ubiquitinated site affects miR-122 levels in these vesicles.

Following the Editorial Suggestions we also have not explored this in detail.

Reviewer 3

While data on the effects of starvation and HuR on miR-122 content of exosomes and cells is acceptable with a few additional controls there are more significant issues with data on ubiquitination of HuR and its binding to miRNA that is used to develop a model of how HuR affects levels of miR-122 in exosomes and cells.

We appreciate the concern of the reviewer and accept the fact that the direct role of HuR ubiquitination in exosomal export of miRNA was less explored in the previous version. In this modified version we have tried to bridge that gap to formulate a HuR driven miRNA export model that is largely supported by the new data incorporated in the revised version.

1. Fig.1A- is there a size change in miR-122 in starved cells? There appears to be a slight downward shift.

We apologise for the confusion what is likely caused by the deformity in the gel (during the run) used for the blotting. We have now introduced a new Northern blot data in Fig. 1A that clearly demonstrate that there is no change in the miRNA size in starved cells. This also rules out the possibility of trimming and trailing of miR-122, for its selection for export, in starved cells. Please also note that the size of miRNA exported out and present in exosome is identical to that detected in the cellular miRNA pool.

2. Provide evidence for specificity of HuR antibody. Blots are often messy.

We are sorry for the same. We have introduced sufficiently larger panels in Fig. 3B and Fig. 2C to confirm the exclusive detection of HuR by the mouse 3A2 anti-HuR monoclonal antibody used throughout this study. Please also note that it also detects bands of predicted size for the HA-tagged full length or truncated versions of the protein apart from the 36 KDa endogenous HuR as shown in

this manuscript. Its specificity is also evident from the Western blot data with HuR IPed and HA-Ago2 IPed materials where it does not detect other major contaminating bands (Fig. EV5B, 6J and 2C). The specificity was further tested when the antibody detects the bacterially expressed His-tagged full length or RRMIII truncated version of HuR but not the other bacterial expressed recombinant proteins (Fig. 5C).

3. Test for levels of CAT1 and aldolase in exosome fractions of starved and fed to see if mRNA comes with miR-122 into exosomes. Mechanistically important.

This was an excellent suggestion! However after measuring the levels of target mRNAs in the exosomes of Fed and Starved cells we could not detect an appreciable level of these mRNAs in exosomal pool. At least they were not in the reliable range of detection. Although GAPDH mRNAs was detectable but it did not show any change with starvation. In the discussion part, we have mentioned the findings.

4. SMPD2 siRNA decreases release of miR-122 release in exosomes by only about 50%. Not all exosome populations are likely affected. In starved cells miR-122 release increases and SMPD2 siRNA brings this back to the same level as in fed cells. This suggests that while not all exosomes released are SMPD2 siRNA dependent (or 50% of miR-122 is not in exosomes), the exosomes released in response to starvation are SMPD2 -dependent. The Nanosight data in Supp. Fig.1 also suggests multiple size peaks in EV preparations. Which subpopulations by size are SMPD2-dependent, and affected by starvation and contain miR-122. Could be lipid particles or other types of vesicles that contain miR-122. Similarly, comparing Fig.3B and E there are large increases in miR-122 in EV in Fig.3B, but a small increase in CD63+ exosomes isolated with beads in 3F, and a small decrease in miR-122 in EV preparations when exosome release is blocked with GW4869. This again suggests that the majority of miR-122 release in EV preps is not in exosomes.

This is an important question but it is difficult to answer in absolute terms. Unlike Starvation or GW4869 treatment that should show effect on all the cells, the siRNA transfection should reduce SMPD2 only in subset of cells leaving other cell unaffected (almost 50% get transfected; as observed by microscopic measurement for a control cy3-labelled siRNA, but data not shown). Therefore the starvation induced increase of miRNA should be ineffective in those cells that are not getting transfected with siSMPD2. Hence, while measuring the exosomal miRNA level in siSMPD2 treated subpopulation the contribution of those non-transfected cells present in the same culture (1.5 fold increase in miR-122 export under starvation) should be negated to effectively measure the reduction in starvation induced miR-122 export in cells where SMPD2 is downregulated. Therefore the presentation of the data we had for the previous version could be confusing. When, it has been normalized to the siControl treated Starved population, that record a reduction to 59% in the total export higher than what observed for non-starved cells. It implies that in non-starved cells there may be alternative mechanism for miRNA export that is not sensitive to SMPD2 while in the starved cells it is primarily the SMPD2 dependent exosome mediated export that also reflected in the data presented in Figure 1G.

Regarding the relative low change in miR-122 levels in CD63 affinity purified exosomes in HA-HuR expressing cells is due to low level of CD63 in HA-HuR expressing cells. The reduction of CD63 could be due to many reasons (as explained in the text) but this should cause less exosomes to get purified with CD63 affinity beads from HA-HuR expressing cells. When normalized against the CD63 level the value looks similar to what observed with ultracentrifugation method. Similarly treatment of GW4869 could not be 100% effective to block the total exosomal export of miR-122. Please also note that the different population of exosomes with different diameter observed in NTA analysis described in Fig EV1 is due to fusion or fission of exosomes in between them in concentrated suspension that resulted in vesicles of double and triple diameter than the mono-exosomes but they are of same density and content. Therefore they may not be biologically different entity. Even with these explanation and quantitative data for Optiprep gradient analysis that showed majority of extracellular miR-122 from starved cells are with CD63 positive EVs, we introduced sentences in the discussion part to address the concern of the Reviewer.

5. Show that SMPD2 siRNA also decreases miR-122 in cells and increases CAT-1.

We guess it should be opposite what the reviewer wanted to see. SMPD2 down regulation should enhance the miR-122 due to blockage in export and should decrease the CAT-1 level! We have done the experiments and introduced the data in Fig 1H.

6. Starvation induces metabolic signaling pathways and autophagy. Autophagy can degrade RNAs. The authors need to show that disappearance of miRNA in starved cells is not due to autophagy using genetic or siRNA tools, not drugs. Since the amount of exosomes released from cells is so small (ug vs. ug) compared to cells, autophagic degradation rather than exosomal release would seem a much more plausible mechanism for decreased miR-122 levels in cells. At minimum the authors should provide RNA copy number data for exosomes vs. total cell RNA to demonstrate that miR-122 amounts found in exosomes could account for those disappearing from cells.

We thank the reviewer to raise this important issue. We have tried to answer it by doing experiments to check the contribution of autophagy in this process. Interestingly we did not get any increased expression of autophagy markers LC3B and p62 and we did not document any increase (rather a decrease; Fig EV2A) in their expression during the amino acid induced stress. The amino acid starvation induced miRNA lowering is also blocked by siSMPD2 or GW4869, the agent that specifically blocks exosomal miRNA export. Taken together, it strongly argues in favour of exosomal export as the major contributor in miRNA lowering in amino acid starved Huh7 cells. We also measured the copy number of the miR-122 in the exosome and in Huh7 cells. The cellular copy numbers of miR-122 drops from 12,000 to 5,000 upon starvation while the increase in exosomal content account for 50% of it. Considering the half life of a miRNA in exosome is about 2 folds less than in cellular level (unpublished data), the exosomal miR-122 increase can account for almost full amount of miR-122 lowered in starved Huh7 cells. We have added this part in the discussion section of the revised manuscript.

7. In 1G gradients data that miR-122 levels increase in exosome fractions of starved cells is very weak - the effect is weak in an experiment with lots of apparent variability. Better evidence is required. Indeed, this suggests there are important amounts of miR-122 in non-exosomal parts of the EV preparations.

We repeated the experiments and from the data obtained, we agree with the reviewer that there are some non-EV associated miRNAs present in both Fed and Starved Huh7 cells supernatant but the changes that we document in CD63 enriched (Exosome Associated) fraction under starvation is more than 20 fold and that itself is about 10 fold excess over non-CD63 fractions in starved cells (Fig. 1J). We hope now it become more convincing to suggest that CD63 positive exosome fraction is the primary carrier of extracellular miR-122 in starved Huh7 cells. But to address this concern we modified our claim in respective section of the text to ensure that the exosomal contribution in EVs isolated from starved Huh7 cells carrying the miR-122 should not be over stated.

8. To show that starvation requires HuR for increased export of miR-122 in EVs the authors need to include data in fed +control siRNA/HuR siRNA, starved +control siRNA/HuR siRNA for miR-122 levels and CAT-1 levels in EVs and cells.

We understand the importance of the suggested experiments and have incorporated the data for EV associated miR-122 in Starved Huh7 cells treated with respective siRNAs (Fig 2D). Change in CAT-1 level in siCon vs. siHuR has already been mentioned in Figure 2E. In Fed cells the CAT-1 is already under repression state (both transcriptionally and post-transcriptionally by miR-122) therefore a further increase in miR-122 content due to HuR depletion does not have an effect on CAT-1 mRNA level (data is not incorporated in the manuscript)

9. Fig.1H. Is there a statistically significant increase in TUNEL positive cells in starved cells. Large positive control bar makes it difficult to see differences in test samples. Stats required.

We have introduced the required statistical analysis.

10. Fig.2A. Blot for HuR is over-exposed. Show lesser exposure of HuR blot and quantify changes in HuR levels in cells and IPs used for RT-qPCR. Changes in HuR levels could account for changes in HuR bound miRNAs in IPs.

We have shown a lesser exposure blot in Fig. 2A and the quantification was already done as suggested. The relative change of HuR for each set in IPed material has been measured and the IgG HC band has been shown that was used for relative quantification that resulted almost identical results to what obtained in calculation with IPed HuR. Therefore we went on with the old HuR normalized data.

11. Purported differences in Fig. 2F are too small to be credibly published without further analyses such as quantifications of eiF2a-P and eiF4ebp1 over multiple experiments or assays of larger effects on downstream effectors. A scrambled anti-miRNA control should be used rather than anti-let-7. At present this data should be eliminated from the paper.

We have replicated the experiments and have presented multiple data to address the concern of the reviewer. We also provided quantitative data for phosphorylated eIF-2 alpha. In the same context we have analyzed the effect on upstream effectors mTOR and p38 to score corresponding changes (Fig. 2F and EV2C). The use of anti let-7a is justified as expression of let-7a is very low compared to other cell types (only 5% in Huh7 cells relative to its expression in non-hepatic cell line HeLa or RAW264.7). Therefore the possible concern of off target effect would be negligible for anti-let-7a oligo. We hope with these changes the data presented in Figure 2F is now acceptable for its incorporation in the revised version.

12. All bar graphs require statistical analysis. These are missing in at least Fig. 2C, 3E, 5E.

Statistical analysis has been performed and introduced for Fig. 2C, 3E and 5E.

13. In all experiments using IPs of proteins followed by measuring RNA levels, blots for the immunoprecipitated proteins are required to show that similar amounts of proteins were immunoprecipitated in each sample. E.g. Fig. 5A, 2A.

We have incorporated the respective western blots and panels of Fig 5A (Fig. EV5A-B now), 2A and others.

14. Fig. 5C. miRNA associates with HuR after incubation with AGO2-miRNA complexes. Is this specific to HuR or would the same result occur with any protein. Try experiment with a control RNA binding protein.

This is an important concern but we have done the same experiments with HuR deletion mutant HuRDIII which is without the RRM III. This mutant has two RNA binding domains RRMI and II but retains the capacity to bind RNAs with AU-rich sequence. Therefore this experiment not only confers the specificity of full length HuR for the miRNA displacement activity but it also strengthened the importance of RRMIII in miRNA binding activity. We have done the RNA EMSA with labelled miR-122 and have documented concentration dependent binding of miR-122 with full length HuR but not with the RRMIII deletion mutant (Fig. 5E). It further supports our claim that HuR RRMIII is specifically involved in binding the miR-122. Both HuR and its truncated mutant retain the capacity to bind TNF-alpha mRNA ARE element encoding RNA (Fig EV5E).

15. Why does levels of CD63 in EV preps consistently decrease in cells transfected with HuR? Fig. 3H, 3D

Interestingly, this is a repeated observation we had with HA- HuR expressing cells. HuR possibly do so by inhibiting CD63 inclusion in exosomes while HA-HuR does not have any major effect on Alix or other exosomal protein levels. It also lowers CD63 cellular levels. We have shown the data in Fig. 3I and commented on the data in the result part.

16. In many places the lack of molecular mass markers and the messy blots with multiple bands makes it very difficult to understand what is happening. This is particularly bad in Fig. 6 on ubiquitination, but also in Fig. 5E IP HuR blot for AGO2. The lower band for AGO2 in the IP on the right side of the figure aligns with a lower band for AGO2. Is this a variant of AGO2? Show molecular weight markers.

We apologise for infrequent relatively inferior quality western blot data shown in the previous version. We have marked the position of the molecular weight markers for majority of the western

blot data to neglect all possible confusions particularly in western blot with multiple bands. We hope these modifications of the new manuscript will make it acceptable for publication.

17. Fig.5C. miRNA associates with HuR after incubation with AGO2-miRNA complexes. Is this specific to HuR or would the same result occur with any protein. HuR needs to be compared to a control RNA binding protein that is not known to bind miRNA.

We found the issue is redundant to what mentioned in point 14. Please see our explanation against point 14 of this Reviewer and our reply also to Reviewer 2 point 4.

*18. Data showing ubiquitination of HuR at MVB is not acceptable. There is no definitive evidence for HuR ubiquitination, let alone its ubiquitination at MVB. The major issues are:
(a) Necessary controls are lacking in most or all of the IPs (control antibody IP, IP of HuR in cells not-transfected with Ub-HA). This renders it impossible to make conclusions from the data. In most or all of the blots it is unclear which bands are HuR alone or HuR-Ub, or some non-specific band or other splicing or modification of HuR. For example, Fig.5D On right, band for Flag-HuR is already detected with ubiquitin antibody. Does this mean all HuR is ubiquitinated? This seems unlikely. Lack of molecular mass markers in Fig.6 makes it very difficult to understand blots in Fig.6 and elsewhere.*

We can understand the importance of the concern and to address this we have done several new experiments to confirm antibody specificity for IP and detection of ubiquitinated forms of HuR in both control and HA-Ub expressing cells (Figs 6B, D and E). Hope these new panels in Fig. 6 have made it more conclusive and they are supporting our claim that HuR do get ubiquitinated and ubiquitinated HuR binds less miRNAs. We have also incorporated the molecular weight markers positions against each WB data for better visualization of the changes.

In Panel 6G (previously 6D) the western blot was done for HuR and it detects the FLAG-HuR in the HuR and HA IPed materials. Please note the ubiquitinated HuR, enriched in the IPed material obtained with anti-HA antibody was detected with HuR specific antibody and is marked by arrow. Please also note this band is absent in the HuR IPed materials in lane next to it. Therefore all FLAG-HuR is not ubiquitinated but only a fraction of it.

(b) One expects a ladder and/or smear for polyubiquitination of a protein, not distinct bands. Fig.6C need to show blots for HuR, not just HA (Ubiquitin) and should show top of gel.

It usually happens for proteins targeted for degradation, that polyubiquitination results in appearance of ladder. Apart from polyubiquitination, limited ubiquitination (mono, di etc.), involved in ESCRT pathway has also been reported for proteins. This helps their compartmentalization to MVBs. From the previous reports on ubiquitination of HuR, it is clear that HuR undergoes a mono, di- or Tri-ubiquitination under specific conditions. That may explain why in our system, we also detect majority of ubiquitinated HuR are mono or limited-ubiquitinated form (Zhou et al. Genes and Dev, 2013). As per the suggestion, we have shown the blots of HuR and full gel picture for ubiquitinated HuR in majority of panels in Fig. 6. Some of the Source files have also been provided.

(c) It is not clear that HuR is ubiquitinated at MVB in normal conditions without over-expression of HA-Ub. The authors should use an antibody to endogenous ubiquitin (without transfecting Ub-HA)

We performed the recommended experiments and have IPed the Endosome/MVB enriched fractions of non-transfected Huh7 cells with anti-HuR specific antibody and detected Ubiquitinated HuR in the IPed materials with anti-Ub antibody (Fig 6D).

19. Role of hinge region of HuR in export of miR-122 in EVs. This could be interesting but controls are missing including validation that HuR is not ubiquitinated in their systems when lacking the hinge region and validation that HuR lacking the hinge retains miR-122 binding capacity.

We have done immunoprecipitation with anti-HA antibody from cell lysate expressing HA-HuR or its mutant version without the Hinge region and western blotted it for HA and Ub to detect the proteins. We detected that only the full length HuR but not the Hinge region deleted version got ubiquitinated. We also measured the associated miRNA levels in HA-IPed materials and document a

similar level of miRNA binding by HuR and its hinge region deleted version. These data is now incorporated in Fig 6J.

Minor Comments

1. For starvation cells are incubated with Hank's with dialyzed FCS. Is this also exosome-depleted by centrifugation. What is the dialysis cut-off? The authors refer to their starvation method as amino-acid starvation. This is inaccurate as cells are starved of many things (Hanks vs. DMEM and dialyzed FCS).

The 3KDa limit was used for dialysis cut-off and the dialyzed FCS was ultra centrifuged to pre-clear the serum derived exosomes. We agree that apart from the amino acids the cells are also starved for few growth factors and others that are absent in dialyzed serum and Hanks solution. But it has been used previously also by others to do the amino acid starvation experiments (*The Journal of Biological Chemistry*, 274, 30424-30432; *Physiol Rep*. 2014 Mar 1; 2(3): e00238).

2. Something strange is happening with miR-21 in cells. Like miR-122 its binding to HuR increases in starved cells and its levels in EVs are affected. However, miR-21 levels in cells are unaffected. Several explanations are possible. The authors should mention this exception.

We are aware of the strange behaviour of the miR-21 and pointed out the possibilities of its regulation at transcription and processing step in the discussion section.

3. Missing blot confirming increased HuR-HA expression in Fig.3H.

We have introduced the blot to show the change in expression level of HuR in cells expressing HA-HuR

2nd Editorial Decision

13 May 2016

Thank you for the submission of your revised manuscript to our journal. We have now received the referee comments that are pasted below.

As you will see, the referees acknowledge that the manuscript has been improved. However, they also all raise a few remaining points that still require attention. I would therefore like to give you the opportunity to fully address all remaining concerns. Please also submit a new point-by-point response to these concerns with the next, final version of your manuscript.

Overstatements and overinterpretations must be avoided; please describe the data as they are and mark interpretations as such.

Please add the missing statistical information, e.g. specify "n" and error bars in the figure legends, which is missing for figures 3E,F, 4F,G, 5D, EV1A,E, EV3C,E,F,G, EV4B,D, EV5D.

REFeree COMMENTS

Referee #1:

The authors have tried to adress the points made, but choose to follow up on only a few.

My main concern remains is that the data on the mechanisms behind miR122 export is not at all conclusive and should be presented as such to the reader. Colectivley the experiments not more than suggest the conclusions and claims made in the abstract, but definitive proof is not provided.

I cannot understand at all why the authors state that the impact of target mRNA levels (i.e.

availability of binding sites and AGO) on miR122 export is 'out of context'?

I agree with the argumentation that because the authors published their data before as 'relative increases' it will be easier to compare these results. However this does not exclude the possibility that representation in another form may be more informative to the reader.

Better insight into the actual levels/cell/exosomes is important for the interpretation and physiological relevance of the data, I think any reader will understand Ct values. This is also the reason why multiple referees asked for actual copy-number values of miR122. Because this was performed, I can accept their response. Nevertheless, the reader should be made aware in the results section that the relative numbers shown by the authors are meaningful, i.e. the Ct values in their experiments are anywhere between 20-33. The latter Ct value (33) representing more or less 10 miRNA copies. Relative differences measured below this will be unreliable.

Referee #2:

The authors, in this revised manuscript, have included better quality western blots as well as additional statistical analyses and data that further support their original conclusions set forth in their work. They have, in general, addressed the majority of my original concerns. There are, nonetheless, some minor concerns that remain from my original review and that should be addressed by the authors prior to the publication of the manuscript in EMBO reports.

1) The authors argue, based on the gel shifts presented in Fig. 5E, that HuR directly associates with miR-122. The data demonstrating this direct binding of HuR to the miRNA is however very weak. Furthermore, additional negative controls should be included in these experiments to support this observation. The authors should therefore demonstrate, in their gel shifts, that HuR does not bind to a mutated miR-122 as well as other small non-coding RNAs such as the U6 snRNA.

2) One of my original concerns with the manuscript was that the authors failed to identify the HuR residue ubiquitinated in these cells. They furthermore failed to assess how a HuR mutant for this site affects miR-122 levels in vesicles. The authors did not assess this concern since the editor concluded that addressing this issue may be beyond the scope of this manuscript. The editor, nonetheless, did indicate that better evidence for HuR ubiquitination must be presented in the manuscript. Indeed, the IP shown in 6B does not convincingly demonstrate that HuR is indeed ubiquitinated. Furthermore, results in 6H do not convincingly demonstrate that the levels of ubiquitinated HuR increase in cells treated with MG132. Better evidence of HuR ubiquitination should therefore be presented in the manuscript. In addition, the authors should assess the effect of treating cells with MG132 on the association of miR-122 with HuR in Figure 6G and, furthermore, on the loading of miR-122 in EVs.

Referee #3:

The authors have addressed the majority of my concerns. Two brief issues should be taken care of prior to publication.

1. Fig 5D,E. Shows lower binding of miR-122 to HuR delta RRMIII vs Full-length HuR, however in the gel accompanying this figure many fold less HuR delta RRMIII is present than Full-length HuR. This could account for most of the difference in pull-down of miR-122 between these two versions of HuR. If recovery of the full length and deletion protein were the same in Fig.5E, the same argument would apply.

2. I agree that protocols for starvation and amino acid starvation vary widely in the literature. This causes much confusion because of apparently conflicting results. The conditions used here should not be described as amino acid starvation as HBSS lacks glucose, many vitamins, and small metabolites and proteins compared to DMEM with dialyzed FBS. An alternative terminology should be used.

Please find the final revised version of the manuscript titled “*Ubiquitination Restricts miRNA Binding of HuR and Augments Extracellular Export of miR-122 in Stressed Hepatic Cells*” that we are submitting for publication in EMBO Reports.

In the accompanying response letter you will find our point-by-point response to the remaining concerns of the Reviewers. In the revised version, we have not only restricted our claim as per the data presented in the paper but also avoided any over statement and over interpretation as such. Apart from providing some clarifications and explanation against the concerns of Reviewers 1 and 3, additional experiments described in Fig. 5, 6 and EV5 to address the concerns of Reviewer 2 have been incorporated. The changed text parts are in “red”.

The missing statistical information for some of the panels, have now been introduced in the respective figure legends as per the suggestions. We have modified the synopsis image by increasing its text size.

We have also included the new authors checklist with specific information related to statistics that was limiting or missing in the originally submitted version.

Response to Reviewers' Comments

Referee #1:

The authors have tried to address the points made, but choose to follow up on only a few.

My main concern remains is that the data on the mechanisms behind miR122 export is not at all conclusive and should be presented as such to the reader. Collectively the experiments not more than suggest the conclusions and claims made in the abstract, but definitive proof is not provided.

We can understand the concern of the reviewer and have paid attention to avoid overstatement or over interpretations of the presented data throughout the text and abstract part of the final revised manuscript. We have strictly focussed on the findings and stated them without any exaggeration.

I cannot understand at all why the authors state that the impact of target mRNA levels (i.e. availability of binding sites and AGO) on miR122 export is 'out of context'?

We apologise for the misinterpretation of the Reviewers' concern, therefore a detailed discussion on the possible important role of miRNA targets in controlling HuR driven export of miRNAs in mammalian cells have now been incorporated. We have clearly mentioned in the discussion that we did not perform any *in vivo* experiment to neglect or establish the phenomenon of target driven inhibition of exosomal miRNA export but only highlighted the data obtained from the *in vitro* HuR binding experiments depicted in Fig EV5F. These data show how the availability of targets with strong miRNA binding sites could neutralize the HuR binding of respective miRNAs. Hence, these targets could potentially influence the export of respective miRNAs in an *in vivo* context.

I agree with the argumentation that because the authors published their data before as 'relative increases' it will be easier to compare these results. However this does not exclude the possibility that representation in an another form may be more informative to the reader.

We also fully agree with the Reviewer that the data representation could have been made in different formats which could be equally representative and understandable. But we would like to request the Reviewer to allow us to keep the present form of data as it is for the sake of the continuity that s/he has also mentioned in her/his response.

Better insight into the actual levels/cell/exosomes is important for the interpretation and physiological relevance of the data, i think any reader will understand Ct values. This is also the reason why multiple referees asked for actual copy-number values of miR122. Because this was performed, i can accept their response. Nevertheless, the reader should be made aware in the results section that the relative numbers shown by the authors are meaningful, i.e. the Ct values in

their experiments are anywhere between 20-33. The latter Ct value (33) representing more or less 10 miRNA copies. Relative differences measured below this will be unreliable.

We agree with the Reviewers' view and have mentioned the Ct values range taken into consideration for each experiment to calculate the relative changes of miRNA levels. These are now either mentioned in the main text or in the legends of the figures.

Referee #2:(Original Response)

The authors, in this revised manuscript, have included better quality western blots as well as additional statistical analyses and data that further support their original conclusions set forth in their work. They have, in general, addressed the majority of my original concerns. There are, nonetheless, some minor concerns that remain from my original review and that should be addressed by the authors prior to the publication of the manuscript in EMBO reports.

1)The authors argue, based on the gel shifts presented in Fig. 5E, that HuR directly associates with miR-122. The data demonstrating this direct binding of HuR to the miRNA is however very weak. Furthermore, additional negative controls should be included in these experiments to support this observation. The authors should therefore demonstrate, in their gel shifts, that HuR does not bind to a mutated miR-122 as well as other small non-coding RNAs such as the U6 snRNA.

The K_d for miR-122 binding to HuR (evident from EMSA) is 50nM, that is comparable to K_d values of miRNA binding to Ago2 [20-80 nM as shown by Lima et al. J Biol Chem (2009) 284:26017–26028 and Tan et al. Nucleic Acids Res (2009) 37:7533–7545]. Therefore, HuR should be able to compete with Ago2 for miRNA binding. Additionally miRNA-HuR binding K_d is in the same range to what has been observed previously for other canonical HuR binding element containing RNAs [Fig. 3 in Kundu et al, Nucl. Acids Res. (2012) 40 (11): 5088-5100]. Evidences for HuR specificity for its substrates are substantial and numerous papers have already been published which confirm that bindings of HuR with its targets are specific. Regarding the specificity of miRNA binding of HuR, Poria et al. has already highlighted the specificity of HuR-miRNA binding through EMSA assay (Fig. 5 in Poria et al. *Oncogene* (2015) 35, 1703–1715). Selectivity of HuR for its target miRNAs also get supported by the other experiments we have done in this manuscript when we measured HuR bound miRNA levels that could vary in different context (IP experiments in Fed and Starved cells, Fig. 2A). Therefore, we think additional experiment may no-longer be necessary to re-confirm the specificity of HuR for its binding to miRNAs.

2)One of my original concerns with the manuscript was that the authors failed to identify the HuR residue ubiquitinated in these cells. They furthermore failed to assess how a HuR mutant for this site affects miR-122 levels in vesicles. The authors did not assess this concern since the editor concluded that addressing this issue may be beyond the scope of this manuscript. The editor, nonetheless, did indicate that better evidence for HuR ubiquitination must be presented in the manuscript. Indeed, the IP shown in 6B does not convincingly demonstrate that HuR is indeed ubiquitinated. Furthermore, results in 6H do not convincingly demonstrate that the levels of ubiquitinated HuR increases in cells treated with MG132. Better evidence of HuR ubiquitination should therefore be presented in the manuscript. In addition, the authors should assess the affect of treating cells with MG132 on the association of miR-122 with HuR in Figure 6G and, furthermore, on the loading of miR-122 in EVs.

The other concern that the Reviewer 2 shared with us is about the HuR ubiquitination. There are at least two additional papers that have already shown context dependent ubiquitination of HuR in mammalian cells (Abdelmohsen et al. EMBO J. 2009;28(9):1271-82 and Zhou et al. Genes & Dev. 2013. 27: 1046-1058). Therefore, we guess there should not be any doubt about the fact that HuR ubiquitination does happen in mammalian cells. Moreover, unlike what Reviewer 2 had pointed out, we have now quantified and observed substantial increase in ubiquitinated HuR content upon MG132 treatment (Fig. 6H, quantification data will be provided in the final version). miRNA binding is primarily by non-ubiquitinated form of HuR (also supported by experiments in Fig. 6J, where ubiquitination defective form of HuR does not lose its binding capacity to miRNA). Ubiquitination releases miRNAs from HuR and MG132 stabilizes ubiquitinated HuR by preventing

the proteosomal degradation of ubiquitinated HuR. Therefore, we could not understand why there should be a change in the levels of miRNA bound to HuR upon MG132 treatment as it specifically affects the stabilization of ubiquitinated pool of HuR which is no-more bound to miRNAs! Furthermore, this concern with MG132 experiment should have been raised during the initial review itself and not during the re-review as this data was already included in the initial version considered for the 1st round of review.

Response from Referee 2 and our reply:

In my opinion the experiments and controls I requested are very important to support the main conclusion of the paper. In fact without these controls the suspension will always be there that some of the main observations of the paper are an artifact. In addition, unlike the authors' claim these are simple and quick experiments that can be done within couple weeks. Below is the reasons for this opinion:

1) The binding of HuR to the miRNA in the EMSA appears visually to be weak. It is difficult to see, for example, an increase in band intensity when increasing concentrations of HuR are incubated with the radioactive miR-122 probe. My concerns, however, would be alleviated by the inclusion of text in the results section explaining that HuR binds to this miRNA with a K_d of 50nM and that these results are similar to those published by others.

As suggested by the Reviewer, we have mentioned the K_d value of HuR binding to miR-122 and introduced the related discussion in the main text as an additional paragraph highlighting the HuR-miRNA binding issues in the discussion part of the revised text. Additionally we have incorporated an additional EMSA data with HuR in Fig.EV5F where the increasing HuR binding of miR-122 with increasing concentrations of HuR is clearly evident.

2) Furthermore, originally I had asked that the authors perform an EMSA with a mutated miR-122 or another small RNA such as U6. The inclusion of these negative controls in their experiments was meant to demonstrate that HuR specifically binds to miR-122. The lack of binding of HuR to these additional negative controls therefore would reinforce the conclusion that HuR specifically binds to the miR-122. The authors argue that the specificity of HuR binding to a miRNA was previously described by Porial et.al. (Oncogene, 2016). Porial et.al demonstrated, however, in their manuscript, that HuR binds to miR-21 by performing EMSA experiments that included 3 negative controls in their experiments including a different miRNA (miR-125), a mutated miR-21 and a non-specific RNA of equivalent length. The IP experiments, as I previously explained, do not demonstrate that HuR directly associates with this miRNA in cells. Based on my arguments stated above I remained convince, as stated in my previous review, that a mutated miR-122 or another nonspecific small RNA (such as U6) should be included in the experiments to strengthen the conclusion that the binding is specific.

The preferential binding of HuR to the miRNAs is mediated by its RRMIII as mutant HuR without the RRM III could not get associated with miR-122 (Fig. 5). This is one aspect of HuR-miRNA binding specificity. Related to specificity for its substrate, HuR preferentially binds to the target mRNAs having AU-rich sequences. But there are also evidences that HuR could also bind the GU-rich sequence (GRE) on a RNA where GRE-binding by the protein HuR itself controls alternative splicing of the mRNA encoding HuR in mammalian cells to control its own expression (Dai et al., Nucleic Acids Res. 2012 Jan; 40(2): 787–800). We found GU-richness in miR-122 and miR-21, the miRNAs that showed preferential association with HuR (Poria et al 2015 and Fig. 5). Therefore, it is possible that RRMIII of HuR that is known to play a role in interaction of poly A stretch on a mRNA, may also interact with miRNAs with GU-rich sequence to select them for export.

As per the suggestions of the Reviewer 2 to include the control RNA in the EMSA assay, we have performed the EMSA assay in presence of miR-122 antisense RNA (miR-122*). The data in Fig.EV5F suggests complete inhibition of miR-122 binding of HuR when the substrate RNA (miR-122*) is present. The double stranded miR122-miR-122* hybrid does not -allow binding with HuR. Therefore HuR binding to the miRNAs should also be affected by the abundance of their substrates. We also do find partial inhibition of miR-122 binding by HuR in presence of TNF-alpha ARE containing RNA having canonical HuR binding sites as a competitor (Fig. 5E). TNF-alpha ARE

RNA should bind HuR thereby reducing the availability of free HuR for miR-122 bindings and reduce HuR bound miR-122 level as evident in this assay.

To approach the specificity issue of HuR binding to different miRNAs, we wanted to test the miRNA binding specificity of HuR in a context when all other competitor miRNAs are present. We have used an alternative approach to do this experiment where we incubated the miRVana kit isolated small RNA pool taken from Huh7 cells and incubated them with recombinant HuR in an *in vitro* binding reaction and subsequently recovered the HuR bound RNAs and quantified them. After amplification, relative Ct values for different HuR-bound miRNAs were found to be different but were not proportional to the contents of respective miRNAs in the input miRNA pool used for binding. As expected, miR-21 and miR-122 showed relatively higher binding with HuR while miR-24 was not detected as HuR associated RNA. The amount of U6 RNA bound to HuR was negligible and was not in the reliable detection level (Ct values >35). This experiment highlights HuR specificity for its target miRNAs and is important to address the concern of the Reviewer related to that.

Interestingly, like HuR substrate miRNAs, the miR-122 antisense RNA miR-122* too contains a stretch of GU rich sequence that binds with HuR but with much less affinity ($K_d=200\text{nM}$) than that of HuR- miR-122 binding ($K_d= 50\text{nM}$). As HuR specifically binds to single stranded RNA and can also bind the RNA containing miR-122* sequence, HuR may play a role in controlling degradation or export of miRNA* strands. Thus, it could potentially regulate miRNA formation in animal cells. This will be an interesting problem for future exploration. We discussed this possibility in the modified discussion part.

From the above experiments, it seems that HuR binding to the miRNAs is controlled by the single stranded property or nature of the RNA. Therefore, within the single stranded RNA pool, the miRNAs with GRE sequences such as miR-122, possibly bind HuR with more affinity. Hence, they are selected for EV-mediated export. These points have been addressed in details in the discussion part of the main text.

3) One of my concerns in my previous review was that the authors did not provide convincing data to suggest that HuR is indeed ubiquitinated in these cells. The authors responded, in their reply to this concern that "...unlike what the Reviewer 2 has pointed out, we have now quantified and observed substantial increase in ubiquitinated HuR content upon MG132 treatment (Fig. 6H, quantification data will be provided in the final version)." It is unclear why these quantifications were not included in the resubmitted manuscript. The inclusion of such quantified data would have surely alleviated my concerns and would have definitely strengthened the conclusion that HuR is ubiquitinated in these cells.

We have included the respective quantification in the appropriate section (Fig.6H) of the result part to strengthen our claim about HuR ubiquitination. We also understand the importance of confirmation related to HuR ubiquitination. To further substantiate our claim regarding ubiquitination of HuR, we have done an *in vitro* ubiquitination assay with recombinant HuR to detect whether the ubiquitination of HuR happens *in vitro*. The ubiquitination assay was performed by incubating recombinant His tagged HuR and HA-Ub expressing Huh7 cell extract in presence of ATP *in vitro*. The western blot data for input and cell extract along with immunoprecipitated materials were used to detect the ubiquitinated HuR formed during the *in vitro* ubiquitination reaction (Fig 6B).

4) Furthermore, during the first review of this manuscript I had asked the authors to assess if a HuR mutant for the ubiquitination site would affect loading of miR-122 in EVs. The authors did not address this issue indicating that they were following editorial suggestions. Despite being in disagreement with the editor's suggestion I had asked the authors to minimally assess these effects in cells treated with MG132. The premise of the experiment was to assess if the binding of HuR to the miR-122 and their loading to EVs in MG132 treated cells would decrease when compared to untreated cells due to the increase in HuR ubiquitinated levels which would result, over time, in the decreased amount of free, unmodified HuR in these cells. Although I remain convinced of the importance of performing such experiments, I leave it up to editor to decide whether they should be included in the manuscript.

Following the Editors' suggestions we have abstained from exploring this point further in this manuscript.

Referee #3:

The authors have addressed the majority of my concerns. Two brief issues should be taken care of prior to publication.

1. Fig 5D,E. Shows lower binding of miR-122 to HuR delta RRMIII vs Full-length HuR, however in the gel accompanying this figure many fold less HuR delta RRMIII is present than Full-length HuR. This could account for most of the difference in pull-down of miR-122 between these two versions of HuR. If recovery of the full length and deletion protein were the same in Fig.5E, the same argument would apply.

We apologise for the confusion. The less recovered level of HuR delta RRMIII in the immunoprecipitated material was indeed taken into account while calculating the amount of miR-122 bound to it. Individually, the amount of recovered RNA was measured and normalized against the amount of HuR and HuR delta RRMIII proteins obtained from corresponding immunoprecipitated materials. This is now clearly mentioned in the figure legend to avoid any further confusion. For the EMSA assay in Fig. 5E the amount of HuR and HuR delta RRMIII were used in molar equivalent levels and were pre-measured from the Coomassie stained gel against BSA standards and concentrations of protein in respective HuR preparations were calculated. Based on that, molar equivalent amount of both proteins were used in the EMSA to compare their binding efficacy for miR-122. This information is now included in the figure legends.

2. I agree that protocols for starvation and amino acid starvation vary widely in the literature. This causes much confusion because of apparently conflicting results. The conditions used here should not be described as amino acid starvation as HBSS lacks glucose, many vitamins, and small metabolites and proteins compared to DMEM with dialyzed FBS. An alternative terminology should be used.

We understand the importance of this concern and have used the term "starved for metabolites including amino acids" or simply "starved" to replace "amino acids starved" whenever applicable. We prefer to retain the term Fed/Starved as they are good representative words to describe the status of the cells that have been used in the assays described.

3rd Editorial Decision

09 June 2016

I am very pleased to accept your manuscript for publication in the next available issue of EMBO reports. Thank you for your contribution to our journal.

Corresponding Author Name: Suvendra N. Bhattacharyya

Manuscript Number: EMBOR-2015-41930V3